# ACTIVATION MAXIMIZATION GENERATIVE ADVERSARIAL NETS

**Zhiming Zhou, Han Cai**
Shanghai Jiao Tong University
heyohai,hcai@apex.sjtu.edu.cn

**Shu Rong**
Yitu Tech
shu.rong@yitu-inc.com

**Yuxuan Song, Kan Ren**
Shanghai Jiao Tong University
songyuxuan,kren@apex.sjtu.edu.cn

**Jun Wang**
University College London
j.wang@cs.ucl.ac.uk

**Weinan Zhang, Yu Yong**
Shanghai Jiao Tong University
wnzhang@sjtu.edu.cn, yyu@apex.sjtu.edu.cn

## ABSTRACT

Class labels have been empirically shown useful in improving the sample quality of generative adversarial nets (GANs). In this paper, we mathematically study the properties of the current variants of GANs that make use of class label information. With class aware gradient and cross-entropy decomposition, we reveal how class labels and associated losses influence GAN's training. Based on that, we propose Activation Maximization Generative Adversarial Networks (AM-GAN) as an advanced solution. Comprehensive experiments have been conducted to validate our analysis and evaluate the effectiveness of our solution, where AM-GAN outperforms other strong baselines and achieves state-of-the-art Inception Score (8.91) on CIFAR-10. In addition, we demonstrate that, with the Inception ImageNet classifier, Inception Score mainly tracks the diversity of the generator, and there is, however, no reliable evidence that it can reflect the true sample quality. We thus propose a new metric, called AM Score, to provide more accurate estimation on the sample quality. Our proposed model also outperforms the baseline methods in the new metric.

## 1 INTRODUCTION

Generative adversarial nets (GANs) (Goodfellow et al., 2014) as a new way for learning generative models, has recently shown promising results in various challenging tasks, such as realistic image generation (Nguyen et al., 2016b; Zhang et al., 2016; Gulrajani et al., 2017), conditional image generation (Huang et al., 2016b; Cao et al., 2017; Isola et al., 2016), image manipulation (Zhu et al., 2016) and text generation (Yu et al., 2016).

Despite the great success, it is still challenging for the current GAN models to produce convincing samples when trained on datasets with high variability, even for image generation with low resolution, e.g., CIFAR-10. Meanwhile, people have empirically found taking advantages of class labels can significantly improve the sample quality.

There are three typical GAN models that make use of the label information: CatGAN (Springenberg, 2015) builds the discriminator as a multi-class classifier; LabelGAN (Salimans et al., 2016) extends the discriminator with one extra class for the generated samples; AC-GAN (Odena et al., 2016) jointly trains the real-fake discriminator and an auxiliary classifier for the specific real classes. By taking the class labels into account, these GAN models show improved generation quality and stability. However, the mechanisms behind them have not been fully explored (Goodfellow, 2016).

In this paper, we mathematically study GAN models with the consideration of class labels. We derive the gradient of the generator's loss w.r.t. class logits in the discriminator, named as class-aware gradient, for LabelGAN (Salimans et al., 2016) and further show its gradient tends to guide each generated sample towards being one of the specific real classes. Moreover, we show that AC-GAN (Odena et al., 2016) can be viewed as a GAN model with hierarchical class discriminator. Based on

the analysis, we reveal some potential issues in the previous methods and accordingly propose a new method to resolve these issues.

Specifically, we argue that a model with explicit target class would provide clearer gradient guidance to the generator than an implicit target class model like that in (Salimans et al., 2016). Comparing with (Odena et al., 2016), we show that introducing the specific real class logits by replacing the overall real class logit in the discriminator usually works better than simply training an auxiliary classifier. We argue that, in (Odena et al., 2016), adversarial training is missing in the auxiliary classifier, which would make the model more likely to suffer mode collapse and produce low quality samples. We also experimentally find that predefined label tends to result in intra-class mode collapse and correspondingly propose dynamic labeling as a solution. The proposed model is named as Activation Maximization Generative Adversarial Networks (AM-GAN). We empirically study the effectiveness of AM-GAN with a set of controlled experiments and the results are consistent with our analysis and, note that, AM-GAN achieves the state-of-the-art Inception Score (8.91) on CIFAR-10.

In addition, through the experiments, we find the commonly used metric needs further investigation. In our paper, we conduct a further study on the widely-used evaluation metric Inception Score (Salimans et al., 2016) and its extended metrics. We show that, with the Inception Model, Inception Score mainly tracks the diversity of generator, while there is no reliable evidence that it can measure the true sample quality. We thus propose a new metric, called AM Score, to provide more accurate estimation on the sample quality as its compensation. In terms of AM Score, our proposed method also outperforms other strong baseline methods.

The rest of this paper is organized as follows. In Section 2, we introduce the notations and formulate the LabelGAN (Salimans et al., 2016) and AC-GAN* (Odena et al., 2016) as our baselines. We then derive the class-aware gradient for LabelGAN, in Section 3, to reveal how class labels help its training. In Section 4, we reveal the overlaid-gradient problem of LabelGAN and propose AM-GAN as a new solution, where we also analyze the properties of AM-GAN and build its connections to related work. In Section 5, we introduce several important extensions, including the dynamic labeling as an alternative of predefined labeling (i.e., class condition), the activation maximization view and a technique for enhancing the AC-GAN*. We study Inception Score in Section 6 and accordingly propose a new metric AM Score. In Section 7, we empirically study AM-GAN and compare it to the baseline models with different metrics. Finally we conclude the paper and discuss the future work in Section 8.

## 2 PRELIMINARIES

In the original GAN formulation (Goodfellow et al., 2014), the loss functions of the generator $G$ and the discriminator $D$ are given as:

$$L_G^{\text{ori}} = -\mathbb{E}_{z \sim p_z(z)}[\log D_r(G(z))] \triangleq -\mathbb{E}_{x \sim G}[\log D_r(x)],$$
$$L_D^{\text{ori}} = -\mathbb{E}_{x \sim p_{\text{data}}}[\log D_r(x)] - \mathbb{E}_{x \sim G}[\log(1 - D_r(x))], \tag{1}$$

where $D$ performs binary classification between the real and the generated samples and $D_r(x)$ represents the probability of the sample $x$ coming from the real data.

### 2.1 LABELGAN

The framework (see Eq. (1)) has been generalized to multi-class case where each sample $x$ has its associated class label $y \in \{1, \ldots, K, K+1\}$, and the $K+1^{\text{th}}$ label corresponds to the generated samples (Salimans et al., 2016). Its loss functions are defined as:

$$L_G^{\text{lab}} = -\mathbb{E}_{x \sim G}[\log \sum_{i=1}^{K} D_i(x)] \triangleq -\mathbb{E}_{x \sim G}[\log D_r(x)], \tag{2}$$

$$L_D^{\text{lab}} = -\mathbb{E}_{(x,y) \sim p_{\text{data}}}[\log D_y(x)] - \mathbb{E}_{x \sim G}[\log D_{K+1}(x)], \tag{3}$$

where $D_i(x)$ denotes the probability of the sample $x$ being class $i$. The loss can be written in the form of cross-entropy, which will simplify our later analysis:

$$L_G^{\text{lab}} = \mathbb{E}_{x \sim G}[H([1, 0], [D_r(x), D_{K+1}(x)])], \tag{4}$$

$$L_D^{\text{lab}} = \mathbb{E}_{(x,y) \sim p_{\text{data}}}[H(v(y), D(x))] + \mathbb{E}_{x \sim G}[H(v(K+1), D(x))], \tag{5}$$

where $D(x) = [D_1(x), D_2(x), ..., D_{K+1}(x)]$ and $v(y) = [v_1(y), \ldots, v_{K+1}(y)]$ with $v_i(y) = 0$ if $i \neq y$ and $v_i(y) = 1$ if $i = y$. $H$ is the cross-entropy, defined as $H(p, q) = -\sum_i p_i \log q_i$. We would refer the above model as LabelGAN (using class labels) throughout this paper.

## 2.2 AC-GAN*

Besides extending the original two-class discriminator as discussed in the above section, Odena et al. (2016) proposed an alternative approach, i.e., AC-GAN, to incorporate class label information, which introduces an auxiliary classifier $C$ for real classes in the original GAN framework. With the core idea unchanged, we define a variant of AC-GAN as the following, and refer it as AC-GAN*:

$$L_G^{\text{ac}}(x, y) = \mathbb{E}_{(x,y)\sim G}\big[H\big([1,0], [D_r(x), D_f(x)]\big)\big] \tag{6}$$

$$+ \mathbb{E}_{(x,y)\sim G}\big[H\big(u(y), C(x)\big)\big], \tag{7}$$

$$L_D^{\text{ac}}(x, y) = \mathbb{E}_{(x,y)\sim p_{\text{data}}}\big[H\big([1,0], [D_r(x), D_f(x)]\big)\big] + \mathbb{E}_{(x,y)\sim G}\big[H\big([0,1], [D_r(x), D_f(x)]\big)\big] \tag{8}$$

$$+ \mathbb{E}_{(x,y)\sim p_{\text{data}}}\big[H\big(u(y), C(x)\big)\big], \tag{9}$$

where $D_r(x)$ and $D_f(x) = 1 - D_r(x)$ are outputs of the binary discriminator which are the same as vanilla GAN, $u(\cdot)$ is the vectorizing operator that is similar to $v(\cdot)$ but defined with $K$ classes, and $C(x)$ is the probability distribution over $K$ real classes given by the auxiliary classifier.

In AC-GAN, each sample has a coupled target class $y$, and a loss on the auxiliary classifier w.r.t. $y$ is added to the generator to leverage the class label information. We refer the losses on the auxiliary classifier, i.e., Eq. (7) and (9), as the auxiliary classifier losses.

The above formulation is a modified version of the original AC-GAN. Specifically, we omit the auxiliary classifier loss $\mathbb{E}_{(x,y)\sim G}[H(u(y), C(x))]$ which encourages the auxiliary classifier $C$ to classify the fake sample $x$ to its target class $y$. Further discussions are provided in Section 5.3. Note that we also adopt the $-\log(D_r(x))$ loss in generator.

## 3 CLASS-AWARE GRADIENT

In this section, we introduce the class-aware gradient, i.e., the gradient of the generator's loss w.r.t. class logits in the discriminator. By analyzing the class-aware gradient of LabelGAN, we find that the gradient tends to refine each sample towards being one of the classes, which sheds some light on how the class label information helps the generator to improve the generation quality. Before delving into the details, we first introduce the following lemma on the gradient properties of the cross-entropy loss to make our analysis clearer.

**Lemma 1.** *With $l$ being the logits vector and $\sigma$ being the softmax function, let $\sigma(l)$ be the current softmax probability distribution and $\hat{p}$ denote the target probability distribution, then*

$$-\frac{\partial H\big(\hat{p}, \sigma(l)\big)}{\partial l} = \hat{p} - \sigma(l). \tag{10}$$

For a generated sample $x$, the loss in LabelGAN is $L_G^{\text{lab}}(x) = H([1,0], [D_r(x), D_{K+1}(x)])$, as defined in Eq. (4). With Lemma 1, the gradient of $L_G^{\text{lab}}(x)$ w.r.t. the logits vector $l(x)$ is given as:

$$-\frac{\partial L_G^{\text{lab}}(x)}{\partial l_k(x)} = -\frac{\partial H\big([1,0], [D_r(x), D_{K+1}(x)]\big)}{\partial l_r(x)}\frac{\partial l_r(x)}{\partial l_k(x)} = \big(1 - D_r(x)\big)\frac{D_k(x)}{D_r(x)}, \quad k \in \{1, \dots, K\},$$

$$-\frac{\partial L_G^{\text{lab}}(x)}{\partial l_{K+1}(x)} = -\frac{\partial H\big([1,0], [D_r(x), D_{K+1}(x)]\big)}{\partial l_{K+1}(x)} = 0 - D_{K+1}(x) = -\big(1 - D_r(x)\big). \tag{11}$$

With the above equations, the gradient of $L_G^{\text{lab}}(x)$ w.r.t. $x$ is:

$$-\frac{\partial L_G^{\text{lab}}(x)}{\partial x} = \sum_{k=1}^{K} -\frac{\partial L_G^{\text{lab}}(x)}{\partial l_k(x)}\frac{\partial l_k(x)}{\partial x} - \frac{\partial L_G^{\text{lab}}(x)}{\partial l_{K+1}(x)}\frac{\partial l_{K+1}(x)}{\partial x}$$

$$= \big(1 - D_r(x)\big)\left(\sum_{k=1}^{K}\frac{D_k(x)}{D_r(x)}\frac{\partial l_k(x)}{\partial x} - \frac{\partial l_{K+1}(x)}{\partial x}\right) = \big(1 - D_r(x)\big)\sum_{k=1}^{K+1}\alpha_k^{\text{lab}}(x)\frac{\partial l_k(x)}{\partial x}, \tag{12}$$

where

$$\alpha_k^{\text{lab}}(x) = \begin{cases} \frac{D_k(x)}{D_r(x)} & k \in \{1, \dots, K\} \\ -1 & k = K+1 \end{cases}. \tag{13}$$

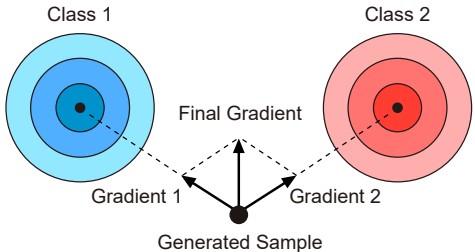

Figure 1: An illustration of the overlaid-gradient problem. When two or more classes are encouraged at the same time, the combined gradient may direct to none of these classes. It could be addressed by assigning each generated sample a specific target class instead of the overall real class.

From the formulation, we find that the overall gradient w.r.t. a generated example $x$ is $1-D_r(x)$, which is the same as that in vanilla GAN (Goodfellow et al., 2014). And the gradient on real classes is further distributed to each specific real class logit $l_k(x)$ according to its current probability ratio $\frac{D_k(x)}{D_r(x)}$.

As such, the gradient naturally takes the label information into consideration: for a generated sample, higher probability of a certain class will lead to a larger step towards the direction of increasing the corresponding confidence for the class. Hence, individually, the gradient from the discriminator for each sample tends to refine it towards being one of the classes in a probabilistic sense.

That is, each sample in LabelGAN is optimized to be one of the real classes, rather than simply to be real as in the vanilla GAN. We thus regard LabelGAN as an implicit target class model. Refining each generated sample towards one of the specific classes would help improve the sample quality. Recall that there are similar inspirations in related work. Denton et al. (2015) showed that the result could be significantly better if GAN is trained with separated classes. And AC-GAN (Odena et al., 2016) introduces an extra loss that forces each sample to fit one class and achieves a better result.

## 4 THE PROPOSED METHOD

In LabelGAN, the generator gets its gradients from the $K$ specific real class logits in discriminator and tends to refine each sample towards being one of the classes. However, LabelGAN actually suffers from the overlaid-gradient problem: all real class logits are encouraged at the same time. Though it tends to make each sample be one of these classes during the training, the gradient of each sample is a weighted averaging over multiple label predictors. As illustrated in Figure 1, the averaged gradient may be towards none of these classes.

In multi-exclusive classes setting, each valid sample should only be classified to one of classes by the discriminator with high confidence. One way to resolve the above problem is to explicitly assign each generated sample a single specific class as its target.

### 4.1 AM-GAN

Assigning each sample a specific target class $y$, the loss functions of the revised-version LabelGAN can be formulated as:

$$L_G^{am} = \mathbb{E}_{(x,y)\sim G}[H(v(y), D(x))], \tag{14}$$

$$L_D^{am} = \mathbb{E}_{(x,y)\sim p_{\text{data}}}[H(v(y), D(x))] + \mathbb{E}_{x\sim G}[H(v(K+1), D(x))], \tag{15}$$

where $v(y)$ is with the same definition as in Section 2.1. The model with aforementioned formulation is named as Activation Maximization Generative Adversarial Networks (AM-GAN) in our paper. And the further interpretation towards naming will be in Section 5.2. The only difference between AM-GAN and LabelGAN lies in the generator's loss function. Each sample in AM-GAN has a specific target class, which resolves the overlaid-gradient problem.

AC-GAN (Odena et al., 2016) also assigns each sample a specific target class, but we will show that the AM-GAN and AC-GAN are substantially different in the following part of this section.

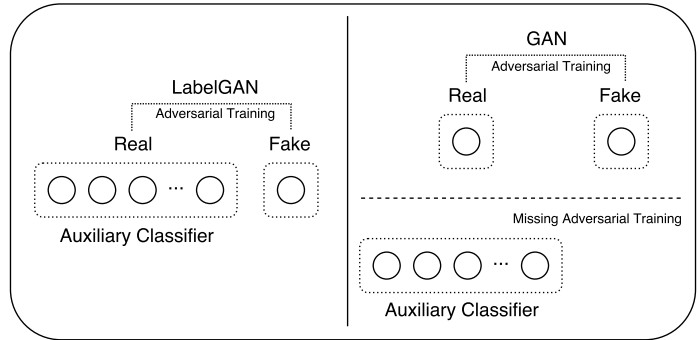

Figure 2: AM-GAN (left) v.s. AC-GAN* (right). AM-GAN can be viewed as a combination of LabelGAN and auxiliary classifier, while AC-GAN* is a combination of vanilla GAN and auxiliary classifier. AM-GAN can naturally conduct adversarial training among all the classes, while in AC-GAN*, adversarial training is only conducted at the real-fake level and missing in the auxiliary classifier.

## 4.2 LABELGAN + AUXILIARY CLASSIFIER

Both LabelGAN and AM-GAN are GAN models with $K+1$ classes. We introduce the following cross-entropy decomposition lemma to build their connections to GAN models with two classes and the $K$-classes models (i.e., the auxiliary classifiers).

**Lemma 2.** *Given* $v = [v_1, \ldots, v_{K+1}]$, $v_{1:K} \triangleq [v_1, \ldots, v_K]$, $v_r \triangleq \sum_{k=1}^{K} v_k$, $R(v) \triangleq v_{1:K}/v_r$ *and* $F(v) \triangleq [v_r, v_{K+1}]$, *let* $\hat{p} = [\hat{p}_1, \ldots, \hat{p}_{K+1}]$, $p = [p_1, \ldots, p_{K+1}]$, *then we have*

$$H\big(\hat{p}, p\big) = \hat{p}_r H\big(R(\hat{p}), R(p)\big) + H\big(F(\hat{p}), F(p)\big). \tag{16}$$

With Lemma 2, the loss function of the generator in AM-GAN can be decomposed as follows:

$$L_G^{\text{am}}(x) = H\big(v(x), D(x)\big) = v_r(x) \cdot \underbrace{H\big(R(v(x)), R(D(x))\big)}_{\text{Auxiliary Classifier G Loss}} + \underbrace{H\big(F(v(x)), F(D(x))\big)}_{\text{LabelGAN G Loss}}. \tag{17}$$

The second term of Eq. (17) actually equals to the loss function of the generator in LabelGAN:

$$H\big(F(v(x)), F(D(x))\big) = H\big([1, 0], [D_r(x), D_{K+1}(x)]\big) = L_G^{\text{lab}}(x). \tag{18}$$

Similar analysis can be adapted to the first term and the discriminator. Note that $v_r(x)$ equals to one. Interestingly, we find by decomposing the AM-GAN losses, AM-GAN can be viewed as a combination of LabelGAN and auxiliary classifier (defined in Section 2.2). From the decomposition perspective, disparate to AM-GAN, AC-GAN is a combination of vanilla GAN and the auxiliary classifier.

The auxiliary classifier loss in Eq. (17) can also be viewed as the cross-entropy version of generator loss in CatGAN: the generator of CatGAN directly optimizes entropy $H(R(D(x)))$ to make each sample have a high confidence of being one of the classes, while AM-GAN achieves this by the first term of its decomposed loss $H(R(v(x)), R(D(x)))$ in terms of cross-entropy with given target distribution. That is, the AM-GAN is the combination of the cross-entropy version of CatGAN and LabelGAN. We extend the discussion between AM-GAN and CatGAN in the Appendix B.

## 4.3 NON-HIERARCHICAL MODEL

With the Lemma 2, we can also reformulate the AC-GAN* as a $K+1$ classes model. Take the generator's loss function as an example:

$$L_G^{\text{ac}}(x, y) = \mathbb{E}_{(x,y)\sim G}\big[H\big([1, 0], [D_r(x), D_f(x)]\big) + H\big(u(y), C(x)\big)\big]$$
$$= \mathbb{E}_{(x,y)\sim G}\big[H\big(v(y), [D_r(x) \cdot C(x), D_f(x)]\big)\big]. \tag{19}$$

In the $K+1$ classes model, the $K+1$ classes distribution is formulated as $[D_r(x) \cdot C(x), D_f(x)]$. AC-GAN introduces the auxiliary classifier in the consideration of leveraging the side information

of class label, it turns out that the formulation of AC-GAN* can be viewed as a hierarchical $K+1$ classes model consists of a two-class discriminator and a $K$-class auxiliary classifier, as illustrated in Figure 2. Conversely, AM-GAN is a non-hierarchical model. All $K+1$ classes stay in the same level of the discriminator in AM-GAN.

In the hierarchical model AC-GAN*, adversarial training is only conducted at the real-fake two-class level, while misses in the auxiliary classifier. Adversarial training is the key to the theoretical guarantee of global convergence $p_G = p_{data}$. Taking the original GAN formulation as an instance, if generated samples collapse to a certain point $x$, i.e., $p_G(x) > p_{data}(x)$, then there must exit another point $x'$ with $p_G(x') < p_{data}(x')$. Given the optimal $D(x) = \frac{p_{data}(x)}{p_G(x)+p_{data}(x)}$, the collapsed point $x$ will get a relatively lower score. And with the existence of higher score points (e.g. $x'$), maximizing the generator's expected score, in theory, has the strength to recover from the mode-collapsed state. In practice, the $p_G$ and $p_{data}$ are usually disjoint (Arjovsky & Bottou, 2017), nevertheless, the general behaviors stay the same: when samples collapse to a certain point, they are more likely to get a relatively lower score from the adversarial network.

Without adversarial training in the auxiliary classifier, a mode-collapsed generator would not get any penalties from the auxiliary classifier loss. In our experiments, we find AC-GAN is more likely to get mode-collapsed, and it was empirically found reducing the weight (such as 0.1 used in Gulrajani et al. (2017)) of the auxiliary classifier losses would help. In Section 5.3, we introduce an extra adversarial training in the auxiliary classifier with which we improve AC-GAN*'s training stability and sample-quality in experiments. On the contrary, AM-GAN, as a non-hierarchical model, can naturally conduct adversarial training among all the class logits.

# 5 EXTENSIONS

## 5.1 DYNAMIC LABELING

In the above section, we simply assume each generated sample has a target class. One possible solution is like AC-GAN (Odena et al., 2016), predefining each sample a class label, which substantially results in a conditional GAN. Actually, we could assign each sample a target class according to its current probability estimated by the discriminator. A natural choice could be the class which is of the maximal probability currently: $y(x) \triangleq \operatorname{argmax}_{i \in \{1,...,K\}} D_i(x)$ for each generated sample $x$. We name this dynamic labeling.

According to our experiments, dynamic labeling brings important improvements to AM-GAN, and is applicable to other models that require target class for each generated sample, e.g. AC-GAN, as an alternative to predefined labeling.

We experimentally find GAN models with pre-assigned class label tend to encounter intra-class mode collapse. In addition, with dynamic labeling, the GAN model remains generating from pure random noises, which has potential benefits, e.g. making smooth interpolation across classes in the latent space practicable.

## 5.2 THE ACTIVATION MAXIMIZATION VIEW

Activation maximization is a technique which is traditionally applied to visualize the neuron(s) of pretrained neural networks (Nguyen et al., 2016a;b; Erhan et al., 2009).

The GAN training can be viewed as an Adversarial Activation Maximization Process. To be more specific, the generator is trained to perform activation maximization for each generated sample on the neuron that represents the log probability of its target class, while the discriminator is trained to distinguish generated samples and prevents them from getting their desired high activation.

It is worth mentioning that the sample that maximizes the activation of one neuron is not necessarily of high quality. Traditionally people introduce various priors to counter the phenomenon (Nguyen et al., 2016a;b). In GAN, the adversarial process of GAN training can detect unrealistic samples and thus ensures the high-activation is achieved by high-quality samples that strongly confuse the discriminator.

We thus name our model the Activation Maximization Generative Adversarial Network (AM-GAN).

### 5.3 AC-GAN[*+]

Experimentally we find AC-GAN easily get mode collapsed and a relatively low weight for the auxiliary classifier term in the generator's loss function would help. In the Section 4.3, we attribute mode collapse to the miss of adversarial training in the auxiliary classifier. From the adversarial activation maximization view: without adversarial training, the auxiliary classifier loss that requires high activation on a certain class, cannot ensure the sample quality.

That is, in AC-GAN, the vanilla GAN loss plays the role for ensuring sample quality and avoiding mode collapse. Here we introduce an extra loss to the auxiliary classifier in AC-GAN[*] to enforce adversarial training and experimentally find it consistently improve the performance:

$$L_D^{\text{ac+}}(x,y) = \mathbb{E}_{(x,y) \sim G}\big[H\big(u(\cdot), C(x)\big)\big],\tag{20}$$

where $u(\cdot)$ represents the uniform distribution, which in spirit is the same as CatGAN (Springenberg, 2015).

Recall that we omit the auxiliary classifier loss $\mathbb{E}_{(x,y) \sim G}\big[H\big(u(y)\big]$ in AC-GAN[*]. According to our experiments, $\mathbb{E}_{(x,y) \sim G}[H(u(y)]$ does improve AC-GAN[*]'s stability and make it less likely to get mode collapse, but it also leads to a worse Inception Score. We will report the detailed results in Section 7. Our understanding on this phenomenon is that: by encouraging the auxiliary classifier also to classify fake samples to their target classes, it actually reduces the auxiliary classifier's ability on providing gradient guidance towards the real classes, and thus also alleviates the conflict between the GAN loss and the auxiliary classifier loss.

## 6 EVALUATION METRICS

One of the difficulties in generative models is the evaluation methodology (Theis et al., 2015). In this section, we conduct both the mathematical and the empirical analysis on the widely-used evaluation metric Inception Score (Salimans et al., 2016) and other relevant metrics. We will show that Inception Score mainly works as a diversity measurement and we propose the AM Score as a compensation to Inception Score for estimating the generated sample quality.

### 6.1 INCEPTION SCORE

As a recently proposed metric for evaluating the performance of generative models, Inception Score has been found well correlated with human evaluation (Salimans et al., 2016), where a publicly-available Inception model $C$ pre-trained on ImageNet is introduced. By applying the Inception model to each generated sample $x$ and getting the corresponding class probability distribution $C(x)$, Inception Score is calculated via

$$\text{Inception Score} = \exp\big(\,\mathbb{E}_x\big[\text{KL}\big(C(x) \parallel \bar{C}^G\big)\big]\,\big),\tag{21}$$

where $\mathbb{E}_x$ is short of $\mathbb{E}_{x \sim G}$ and $\bar{C}^G = \mathbb{E}_x[C(x)]$ is the overall probability distribution of the generated samples over classes, which is judged by $C$, and KL denotes the Kullback-Leibler divergence. As proved in Appendix D, $\mathbb{E}_x\big[\text{KL}\big(C(x) \parallel \bar{C}^G\big)\big]$ can be decomposed into two terms in entropy:

$$\mathbb{E}_x\big[\text{KL}(C(x) \parallel \bar{C}^G)\big] = H(\bar{C}^G) + (-\mathbb{E}_x\big[H\big(C(x)\big)\big]).\tag{22}$$

### 6.2 THE PROPERTIES OF INCEPTION MODEL

A common understanding of how Inception Score works lies in that a high score in the first term $H(\bar{C}^G)$ indicates the generated samples have high diversity (the overall class probability distribution evenly distributed), and a high score in the second term $-\mathbb{E}_x[H(C(x))]$ indicates that each individual sample has high quality (each generated sample's class probability distribution is sharp, i.e., it can be classified into one of the real classes with high confidence) (Salimans et al., 2016).

However, taking CIFAR-10 as an illustration, the data are not evenly distributed over the classes under the Inception model trained on ImageNet, which is presented in Figure 4a. It makes Inception Score problematic in the view of the decomposed scores, i.e., $H(\bar{C}^G)$ and $-\mathbb{E}_x[H(C(x))]$. Such as that one would ask whether a higher $H(\bar{C}^G)$ indicates a better mode coverage and whether a smaller $H(C(x))$ indicates a better sample quality.

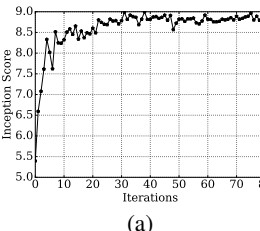 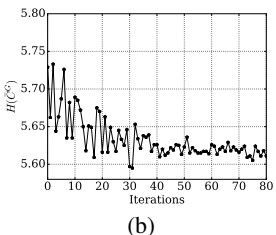 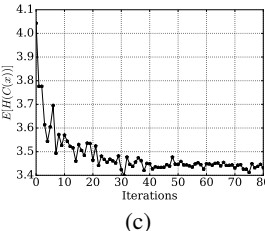

(a) (b) (c)

Figure 3: Training curves of Inception Score and its decomposed terms. a) Inception Score, i.e. $\exp(H(\bar{C}^G) - \mathbb{E}_x[H(C(x))])$; b) $H(\bar{C}^G)$; c) $\mathbb{E}_x[H(C(x))]$. A common understanding of Inception Score is that: the value of $H(\bar{C}^G)$ measures the diversity of generated samples and is expected to increase in the training process. However, it usually tends to decrease in practice as illustrated in (c).

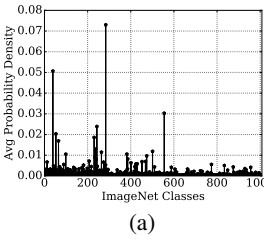 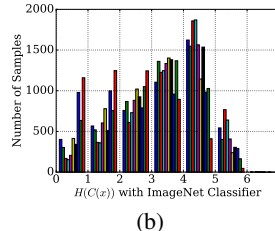 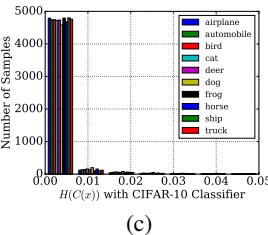

(a) (b) (c)

Figure 4: Statistics of the CIFAR-10 training images. a) $\bar{C}^G$ over ImageNet classes; b) $H(C(x))$ distribution with ImageNet classifier of each class; c) $H(C(x))$ distribution with CIFAR-10 classifier of each class. With the Inception model, the value of $H(C(x))$ score of CIFAR-10 training data is variant, which means, even in real data, it would still strongly prefer some samples than some others. $H(C(x))$ on a classifier that pre-trained on CIFAR-10 has low values for all CIFAR-10 training data and thus can be used as an indicator of sample quality.

We experimentally find that, as in Figure 3b, the value of $H(\bar{C}^G)$ is usually going down during the training process, however, which is expected to increase. And when we delve into the detail of $H(C(x))$ for each specific sample in the training data, we find the value of $H(C(x))$ score is also variant, as illustrated in Figure 4b, which means, even in real data, it would still strongly prefer some samples than some others. The $\exp$ operator in Inception Score and the large variance of the value of $H(C(x))$ aggravate the phenomenon. We also observe the preference on the class level in Figure 4b, e.g., $\mathbb{E}_x[H(C(x))]$=2.14 for trucks, while $\mathbb{E}_x[H(C(x))]$=3.80 for birds.

It seems, for an ImageNet Classifier, both the two indicators of Inception Score cannot work correctly. Next we will show that Inception Score actually works as a diversity measurement.

## 6.3 INCEPTION SCORE AS A DIVERSITY MEASUREMENT

Since the two individual indicators are strongly correlated, here we go back to Inception Score's original formulation $\mathbb{E}_x[\mathrm{KL}(C(x) \| \bar{C}^G)]$. In this form, we could interpret Inception Score as that it requires each sample's distribution $C(x)$ highly different from the overall distribution of the generator $\bar{C}^G$, which indicates a good diversity over the generated samples.

As is empirically observed, a mode-collapsed generator usually gets a low Inception Score. In an extreme case, assuming all the generated samples collapse to a single point, then $C(x)=C^G$ and we would get the minimal Inception Score 1.0, which is the $\exp$ result of zero. To simulate mode collapse in a more complicated case, we design synthetic experiments as following: given a set of $N$ points $\{x_0, x_1, x_2, ..., x_{N-1}\}$, with each point $x_i$ adopting the distribution $C(x_i) = v(i)$ and representing class $i$, where $v(i)$ is the vectorization operator of length $N$, as defined in Section 2.1, we randomly drop $m$ points, evaluate $\mathbb{E}_x[\mathrm{KL}(C(x) \| \bar{C}^G)]$ and draw the curve. As is showed in Figure 5, when $N-m$ increases, the value of $\mathbb{E}_x[\mathrm{KL}(C(x) \| \bar{C}^G)]$ monotonically increases in general, which means that it can well capture the mode dropping and the diversity of the generated distributions.

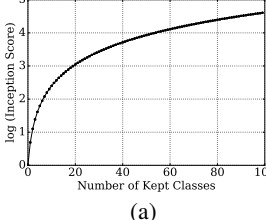
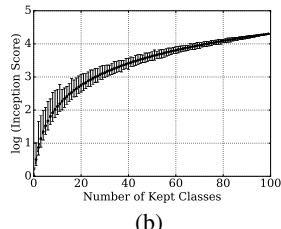

(a)                                                    (b)

Figure 5: Mode dropping analysis of Inception Score. a) Uniform density over classes; b) Gaussian density over classes. The value of $\mathbb{E}_x[\mathrm{KL}(C(x) \parallel \bar{C}^G)]$ monotonically increases in general as the number of kept classes increases, which illustrates Inception Score is able to capture the mode dropping and the diversity of the generated distributions. The error bar indicates the min and max values in 1000 random dropping.

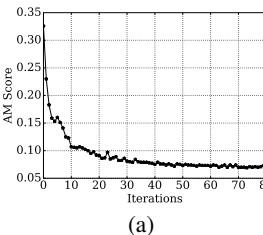
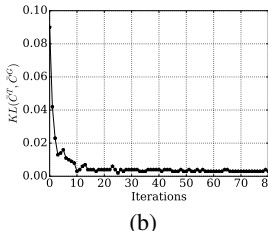
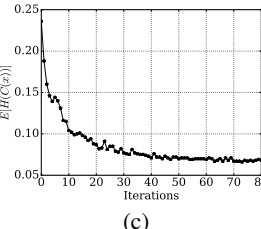

(a)                                    (b)                                    (c)

Figure 6: Training curves of AM Score and its decomposed terms. a) AM Score, i.e. $\mathrm{KL}(\bar{C}^{\mathrm{train}}, \bar{C}^G) + \mathbb{E}_x[H(C(x))]$; b) $\mathrm{KL}(\bar{C}^{\mathrm{train}}, \bar{C}^G)$; c) $\mathbb{E}_x[H(C(x))]$. All of them works properly (going down) in the training process.

One remaining question is that whether good mode coverage and sample diversity mean high quality of the generated samples. From the above analysis, we do not find any evidence. A possible explanation is that, in practice, sample diversity is usually well correlated with the sample quality.

## 6.4 AM SCORE WITH ACCORDINGLY PRETRAINED CLASSIFIER

Note that if each point $x_i$ has multiple variants such as $x_i^1, x_i^2, x_i^3$, one of the situations, where $x_i^2$ and $x_i^3$ are missing and only $x^1$ is generated, cannot be detected by $\mathbb{E}_x[\mathrm{KL}(C(x) \parallel \bar{C}^G)]$ score. It means that with an accordingly pretrained classifier, $\mathbb{E}_x[\mathrm{KL}(C(x) \parallel \bar{C}^G)]$ score cannot detect intra-class level mode collapse. This also explains why the Inception Network on ImageNet could be a good candidate $C$ for CIFAR-10. Exploring the optimal $C$ is a challenge problem and we shall leave it as a future work.

However, there is no evidence that using an Inception Network trained on ImageNet can accurately measure the sample quality, as shown in Section 6.2. To compensate Inception Score, we propose to introduce an extra assessment using an accordingly pretrained classifier. In the accordingly pretrained classifier, most real samples share similar $H(C(x))$ and 99.6% samples hold scores less than 0.05 as showed in Figure 4c, which demonstrates that $H(C(x))$ of the classifier can be used as an indicator of sample quality.

The entropy term on $\bar{C}^G$ is actually problematic when training data is not evenly distributed over classes, for that $\mathrm{argmin}\ H(\bar{C}^G)$ is a uniform distribution. To take the $\bar{C}^{\mathrm{train}}$ into account, we replace $H(\bar{C}^G)$ with a KL divergence between $\bar{C}^{\mathrm{train}}$ and $\bar{C}^G$. So that

$$\text{AM Score} \triangleq \mathrm{KL}(\bar{C}^{\mathrm{train}}, \bar{C}^G) + \mathbb{E}_x\big[H\big(C(x)\big)\big], \qquad (23)$$

which requires $\bar{C}^G$ close to $\bar{C}^{\mathrm{train}}$ and each sample $x$ has a low entropy $C(x)$. The minimal value of AM Score is zero, and the smaller value, the better. A sample training curve of AM Score is showed in Figure 6, where all indicators in AM Score work as expected. [1]

---

[1] Inception Score and AM Score measure the diversity and quality of generated samples, while FID (Heusel et al., 2017) measures the distance between the generated distribution and the real distribution.

| Model | Inception Score | | | | AM Score | | | |
|---|---|---|---|---|---|---|---|---|
| | CIFAR-10 | | Tiny ImageNet | | CIFAR-10 | | Tiny ImageNet | |
| | dynamic | predefined | dynamic | predefined | dynamic | predefined | dynamic | predefined |
| GAN | $7.04 \pm 0.06$ | $7.27 \pm 0.07$ | - | - | $0.45 \pm 0.00$ | $0.43 \pm 0.00$ | - | - |
| GAN* | $7.25 \pm 0.07$ | $7.31 \pm 0.10$ | - | - | $0.40 \pm 0.00$ | $0.41 \pm 0.00$ | - | - |
| AC-GAN* | $7.41 \pm 0.09$ | $7.79 \pm 0.08$ | $7.28 \pm 0.07$ | $7.89 \pm 0.11$ | $0.17 \pm 0.00$ | $0.16 \pm 0.00$ | $1.64 \pm 0.02$ | $1.01 \pm 0.01$ |
| AC-GAN*+ | $8.56 \pm 0.11$ | $8.01 \pm 0.09$ | $10.25 \pm 0.14$ | $8.23 \pm 0.10$ | $0.10 \pm 0.00$ | $0.14 \pm 0.00$ | $1.04 \pm 0.01$ | $1.20 \pm 0.01$ |
| LabelGAN | $8.63 \pm 0.08$ | $7.88 \pm 0.07$ | $10.82 \pm 0.16$ | $8.62 \pm 0.11$ | $0.13 \pm 0.00$ | $0.25 \pm 0.00$ | $1.11 \pm 0.01$ | $1.37 \pm 0.01$ |
| AM-GAN | $\mathbf{8.83 \pm 0.09}$ | $\mathbf{8.35 \pm 0.12}$ | $\mathbf{11.45 \pm 0.15}$ | $\mathbf{9.55 \pm 0.11}$ | $\mathbf{0.08 \pm 0.00}$ | $\mathbf{0.05 \pm 0.00}$ | $\mathbf{0.88 \pm 0.01}$ | $\mathbf{0.61 \pm 0.01}$ |

Table 1: Inception Score and AM Score Results. Models in the same column share the same network structures & hyper-parameters. We applied dynamic / predefined labeling for models that require target classes.

| | AC-GAN* | AC-GAN*+ | LabelGAN | AM-GAN |
|---|---|---|---|---|
| dynamic | **0.61** | 0.39 | 0.35 | 0.36 |
| predefined | 0.35 | 0.36 | 0.32 | 0.36 |

Table 2: The maximum value of mean MS-SSIM of various models over the ten classes on CIFAR-10. High-value indicates obvious intra-class mode collapse. Please refer to the Figure 11 in the Appendix for the visual results.

## 7 EXPERIMENTS

To empirically validate our analysis and the effectiveness of the proposed method, we conduct experiments on the image benchmark datasets including CIFAR-10 and Tiny-ImageNet[2] which comprises 200 classes with 500 training images per class. For evaluation, several metrics are used throughout our experiments, including Inception Score with the ImageNet classifier, AM Score with a corresponding pretrained classifier for each dataset, which is a DenseNet (Huang et al., 2016a) model. We also follow Odena et al. (2016) and use the mean MS-SSIM (Wang et al., 2004) of randomly chosen pairs of images within a given class, as a coarse detector of intra-class mode collapse.

A modified DCGAN structure, as listed in the Appendix F, is used in experiments. Visual results of various models are provided in the Appendix considering the page limit, such as Figure 9, etc. The repeatable experiment code is published for further research[3].

### 7.1 EXPERIMENTS ON CIFAR-10

#### 7.1.1 GAN WITH AUXILIARY CLASSIFIER

The first question is whether training an auxiliary classifier without introducing correlated losses to the generator would help improve the sample quality. In other words, with the generator only with the GAN loss in the AC-GAN* setting. (referring as GAN*)

As is shown in Table 1, it improves GAN's sample quality, but the improvement is limited comparing to the other methods. It indicates that introduction of correlated loss plays an essential role in the remarkable improvement of GAN training.

#### 7.1.2 COMPARISON AMONG DIFFERENT MODELS

The usage of the predefined label would make the GAN model transform to its conditional version, which is substantially disparate with generating samples from pure random noises. In this experiment, we use dynamic labeling for AC-GAN*, AC-GAN*+ and AM-GAN to seek for a fair comparison among different discriminator models, including LabelGAN and GAN. We keep the network structure and hyper-parameters the same for different models, only difference lies in the output layer of the discriminator, i.e., the number of class logits, which is necessarily different across models.

As is shown in Table 1, AC-GAN* achieves improved sample quality over vanilla GAN, but sustains mode collapse indicated by the value 0.61 in MS-SSIM as in Table 2. By introducing adversarial

---

[2]https://tiny-imagenet.herokuapp.com/

[3]Link for anonymous experiment code: https://github.com/ZhimingZhou/AM-GAN

| Model | Score $\pm$ Std. |
|---|---|
| DFM (Warde-Farley & Bengio, 2017) | $7.72 \pm 0.13$ |
| Improved GAN (Salimans et al., 2016) | $8.09 \pm 0.07$ |
| AC-GAN (Odena et al., 2016) | $8.25 \pm 0.07$ |
| WGAN-GP + AC (Gulrajani et al., 2017) | $8.42 \pm 0.10$ |
| SGAN (Huang et al., 2016b) | $8.59 \pm 0.12$ |
| AM-GAN (our work) | $\mathbf{8.91 \pm 0.11}$ |
| Splitting GAN (Guillermo et al., 2017) | $8.87 \pm 0.09$ |
| Real data | $11.24 \pm 0.12$ |

Table 3: Inception Score comparison on CIFAR-10. Splitting GAN uses the class splitting technique to enhance the class label information, which is orthogonal to AM-GAN.

training in the auxiliary classifier, AC-GAN$^{*+}$ outperforms AC-GAN$^*$. As an implicit target class model, LabelGAN suffers from the overlaid-gradient problem and achieves a relatively higher per sample entropy (0.124) in the AM Score, comparing to explicit target class model AM-GAN (0.079) and AC-GAN$^{*+}$ (0.102). In the table, our proposed AM-GAN model reaches the best scores against these baselines.

We also test AC-GAN$^*$ with decreased weight on auxiliary classifier losses in the generator ($\frac{1}{10}$ relative to the GAN loss). It achieves 7.19 in Inception Score, 0.23 in AM Score and 0.35 in MS-SSIM. The 0.35 in MS-SSIM indicates there is no obvious mode collapse, which also conform with our above analysis.

### 7.1.3 INCEPTION SCORE COMPARING WITH RELATED WORK

AM-GAN achieves Inception Score 8.83 in the previous experiments, which significantly outperforms the baseline models in both our implementation and their reported scores as in Table 3. By further enhancing the discriminator with more filters in each layer, AM-GAN also outperforms the orthogonal work (Guillermo et al., 2017) that enhances the class label information via class splitting. As the result, AM-GAN achieves the state-of-the-art Inception Score 8.91 on CIFAR-10.

### 7.1.4 DYNAMIC LABELING AND CLASS CONDITION

It's found in our experiments that GAN models with class condition (predefined labeling) tend to encounter intra-class mode collapse (ignoring the noise), which is obvious at the very beginning of GAN training and gets exasperated during the process.

In the training process of GAN, it is important to ensure a balance between the generator and the discriminator. With the same generator's network structures and switching from dynamic labeling to class condition, we find it hard to hold a good balance between the generator and the discriminator: to avoid the initial intra-class mode collapse, the discriminator need to be very powerful; however, it usually turns out the discriminator is too powerful to provide suitable gradients for the generator and results in poor sample quality.

Nevertheless, we find a suitable discriminator and conduct a set of comparisons with it. The results can be found in Table 1. The general conclusion is similar to the above, AC-GAN$^{*+}$ still outperforms AC-GAN$^*$ and our AM-GAN reaches the best performance. It's worth noticing that the AC-GAN$^*$ does not suffer from mode collapse in this setting.

In the class conditional version, although with fine-tuned parameters, Inception Score is still relatively low. The explanation could be that, in the class conditional version, the sample diversity still tends to decrease, even with a relatively powerful discriminator. With slight intra-class mode collapse, the per-sample-quality tends to improve, which results in a lower AM Score. A supplementary evidence, not very strict, of partial mode collapse in the experiments is that: the $\sum |\frac{\partial G(z)}{\partial z}|$ is around 45.0 in dynamic labeling setting, while it is 25.0 in the conditional version.

The LabelGAN does not need explicit labels and the model is the same in the two experiment settings. But please note that both Inception Score and the AM Score get worse in the conditional version. The only difference is that the discriminator becomes more powerful with an extended layer, which attests

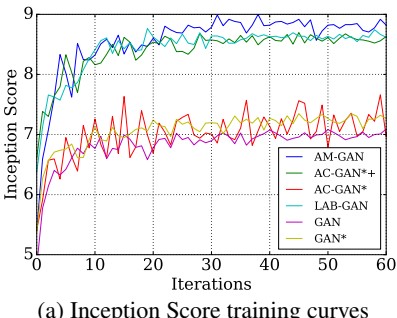 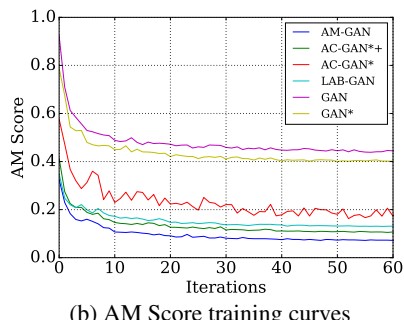

(a) Inception Score training curves       (b) AM Score training curves

Figure 7: The training curves of different models in the dynamic labeling setting.

that the balance between the generator and discriminator is crucial. We find that, without the concern of intra-class mode collapse, using the dynamic labeling makes the balance between generator and discriminator much easier.

### 7.1.5 THE $\mathbb{E}_{(x,y)\sim G}[H(u(y), C(x))]$ LOSS

Note that we report results of the modified version of AC-GAN, i.e., AC-GAN* in Table 1. If we take the omitted loss $\mathbb{E}_{(x,y)\sim G}[H(u(y), C(x))]$ back to AC-GAN*, which leads to the original AC-GAN (see Section 2.2), it turns out to achieve worse results on both Inception Score and AM Score on CIFAR-10, though dismisses mode collapse. Specifically, in dynamic labeling setting, Inception Score decreases from $7.41$ to $6.48$ and the AM Score increases from $0.17$ to $0.43$, while in predefined class setting, Inception Score decreases from $7.79$ to $7.66$ and the AM Score increases from $0.16$ to $0.20$.

This performance drop might be because we use different network architectures and hyper-parameters from AC-GAN (Odena et al., 2016). But we still fail to achieve its report Inception Score, i.e., $8.25$, on CIFAR-10 when using the reported hyper-parameters in the original paper. Since they do not publicize the code, we suppose there might be some unreported details that result in the performance gap. We would leave further studies in future work.

### 7.1.6 THE LEARNING PROPERTY

We plot the training curve in terms of Inception Score and AM Score in Figure 7. Inception Score and AM Score are evaluated with the same number of samples $50k$, which is the same as Salimans et al. (2016). Comparing with Inception Score, AM Score is more stable in general. With more samples, Inception Score would be more stable, however the evaluation of Inception Score is relatively costly. A better alternative of the Inception Model could help solve this problem.

The AC-GAN*'s curves appear stronger jitter relative to the others. It might relate to the counteract between the auxiliary classifier loss and the GAN loss in the generator. Another observation is that the AM-GAN in terms of Inception Score is comparable with LabelGAN and AC-GAN*+ at the beginning, while in terms of AM Score, they are quite distinguishable from each other.

### 7.2 EXPERIMENTS ON TINY-IMAGENET

In the CIFAR-10 experiments, the results are consistent with our analysis and the proposed method outperforms these strong baselines. We demonstrate that the conclusions can be generalized with experiments in another dataset Tiny-ImageNet.

The Tiny-ImageNet consists with more classes and fewer samples for each class than CIFAR-10, which should be more challenging. We downsize Tiny-ImageNet samples from $64 \times 64$ to $32 \times 32$ and simply leverage the same network structure that used in CIFAR-10, and the experiment result is showed also in Table 1. From the comparison, AM-GAN still outperforms other methods remarkably. And the AC-GAN*+ gains better performance than AC-GAN*.

## 8 CONCLUSION

In this paper, we analyze current GAN models that incorporate class label information. Our analysis shows that: LabelGAN works as an implicit target class model, however it suffers from the overlaid-gradient problem at the meantime, and explicit target class would solve this problem. We demonstrate that introducing the class logits in a non-hierarchical way, i.e., replacing the overall real class logit in the discriminator with the specific real class logits, usually works better than simply supplementing an auxiliary classifier, where we provide an activation maximization view for GAN training and highlight the importance of adversarial training. In addition, according to our experiments, predefined labeling tends to lead to intra-class mode collapsed, and we propose dynamic labeling as an alternative. Our extensive experiments on benchmarking datasets validate our analysis and demonstrate our proposed AM-GAN's superior performance against strong baselines. Moreover, we delve deep into the widely-used evaluation metric Inception Score, reveal that it mainly works as a diversity measurement. And we also propose AM Score as a compensation to more accurately estimate the sample quality.

In this paper, we focus on the generator and its sample quality, while some related work focuses on the discriminator and semi-supervised learning. For future work, we would like to conduct empirical studies on discriminator learning and semi-supervised learning. We extend AM-GAN to unlabeled data in the Appendix C, where unsupervised and semi-supervised is accessible in the framework of AM-GAN. The classifier-based evaluation metric might encounter the problem related to adversarial samples, which requires further study. Combining AM-GAN with Integral Probability Metric based GAN models such as Wasserstein GAN (Arjovsky et al., 2017) could also be a promising direction since it is orthogonal to our work.

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

# A  GRADIENT VANISHING & $-\log(D_r(x))$ & LABEL SMOOTHING

## A.1  LABEL SMOOTHING

Label smoothing that avoiding extreme logits value was showed to be a good regularization (Szegedy et al., 2016). A general version of label smoothing could be: modifying the target probability of discriminator)

$$\left[\hat{D}_r(x), \hat{D}_f(x)\right] = \begin{cases} [\lambda_1, 1 - \lambda_1] & x \sim G \\ [1 - \lambda_2, \lambda_2] & x \sim p_{\text{data}} \end{cases}. \tag{24}$$

Salimans et al. (2016) proposed to use only one-side label smoothing. That is, to only apply label smoothing for real samples: $\lambda_1 = 0$ and $\lambda_2 > 0$. The reasoning of one-side label smoothing is applying label smoothing on fake samples will lead to fake mode on data distribution, which is too obscure.

We will next show the exact problems when applying label smoothing to fake samples along with the $\log(1 - D_r(x))$ generator loss, in the view of gradient w.r.t. class logit, i.e., the class-aware gradient, and we will also show that the problem does not exist when using the $-\log(D_r(x))$ generator loss.

## A.2  THE $\log(1 - D_r(x))$ GENERATOR LOSS

The $\log(1 - D_r(x))$ generator loss with label smoothing in terms of cross-entropy is

$$L_G^{\log(1\text{-}D)} = -\mathbb{E}_{x \sim G}\left[H\left([\lambda_1, 1 - \lambda_1], [D_r(x), D_{K+1}(x)]\right)\right], \tag{25}$$

with lemma 1, its negative gradient is

$$-\frac{\partial L_G^{\log(1\text{-}D)}(x)}{\partial l_r(x)} = D_r(x) - \lambda_1, \tag{26}$$

$$\begin{cases} D_r(x) = \lambda_1 & \text{gradient vanishing} \\ D_r(x) < \lambda_1 & D_r(x) \text{ is optimized towards } 0 \\ D_r(x) > \lambda_1 & D_r(x) \text{ is optimized towards } 1 \end{cases}. \tag{27}$$

Gradient vanishing is a well know training problem of GAN. Optimizing $D_r(x)$ towards 0 or 1 is also not what desired, because the discriminator is mapping real samples to the distribution with $D_r(x) = 1 - \lambda_2$.

## A.3  THE $-\log(D_r(x))$ GENERATOR LOSS

The $-\log(D_r(x))$ generator loss with target $[1 - \lambda, \lambda]$ in terms of cross-entropy is

$$L_G^{\text{-}\log(D)} = \mathbb{E}_{x \sim G}\left[H\left([1 - \lambda, \lambda], [D_r(x), D_{K+1}(x)]\right)\right], \tag{28}$$

the negative gradient of which is

$$-\frac{\partial L_G^{\text{-}\log(D)}(x)}{\partial l_r(x)} = (1 - \lambda) - D_r(x), \tag{29}$$

$$\begin{cases} D_r(x) = 1 - \lambda & \text{stationary point} \\ D_r(x) < 1 - \lambda & D_r(x) \text{ towards } 1 - \lambda \\ D_r(x) > 1 - \lambda & D_r(x) \text{ towards } 1 - \lambda \end{cases}. \tag{30}$$

Without label smooth $\lambda$, the $-\log(D_r(x))$ always* preserves the same gradient direction as $\log(1 - D_r(x))$ though giving a difference gradient scale. We must note that non-zero gradient does not mean that the gradient is efficient or valid.

The both-side label smoothed version has a strong connection to Least-Square GAN (Mao et al., 2016): with the fake logit fixed to zero, the discriminator maps real to $\alpha$ on the real logit and maps fake to $\beta$ on the real logit, the generator in contrast tries to map fake sample to $\alpha$. Their gradient on the logit are also similar.

## B CATGAN

The auxiliary classifier loss of AM-GAN can also be viewed as the cross-entropy version of CatGAN: generator of CatGAN directly optimizes entropy $H(R(D(x)))$ to make each sample be one class, while AM-GAN achieves this by the first term of its decomposed loss $H(R(v(x)), R(D(x)))$ in terms of cross-entropy with given target distribution. That is, the AM-GAN is the cross-entropy version of CatGAN that is combined with LabelGAN by introducing an additional fake class.

### B.1 DISCRIMINATOR LOSS ON FAKE SAMPLE

The discriminator of CatGAN maximizes the prediction entropy of each fake sample:

$$L_D^{\text{Cat'}} = \mathbb{E}_{x \sim G}\big[ - H\big(D(x)\big)\big]. \tag{31}$$

In AM-GAN, as we have an extra class on fake, we can achieve this in a simpler manner by minimizing the probability on real logits.

$$L_D^{\text{AM'}} = \mathbb{E}_{x \sim G}\big[H\big(F(v(K{+}1)), F(D(x)))\big)\big]. \tag{32}$$

If $v_r(K{+}1)$ is not zero, that is, when we did negative label smoothing Salimans et al. (2016), we could define $R(v(K{+}1))$ to be a uniform distribution.

$$L_D^{\text{AM''}} = \mathbb{E}_{x \sim G}\big[H\big(R(v(K{+}1)), R(D(x)))\big)\big] \times v_r(K{+}1). \tag{33}$$

As a result, the label smoothing part probability will be required to be uniformly distributed, similar to CatGAN.

## C UNLABELED DATA

In this section, we extend AM-GAN to unlabeled data. Our solution is analogous to CatGAN Springenberg (2015).

### C.1 SEMI-SUPERVISED SETTING

Under semi-supervised setting, we can add the following loss to the original solution to integrate the unlabeled data (with the distribution denoted as $p_{\text{unl}}(x)$):

$$L_D^{\text{unl'}} = \mathbb{E}_{x \sim p_{\text{unl}}}\big[H\big(v(x), D(x)\big)\big]. \tag{34}$$

### C.2 UNSUPERVISED SETTING

Under unsupervised setting, we need to introduce one extra loss, analogy to categorical GAN Springenberg (2015):

$$L_D^{\text{unl''}} = H\big(p_{\text{ref}}, R(\mathbb{E}_{x \sim p_{\text{unl}}}[D(x)])\big), \tag{35}$$

where the $p_{\text{ref}}$ is a reference label distribution for the prediction on unsupervised data. For example, $p_{\text{ref}}$ could be set as a uniform distribution, which requires the unlabeled data to make use of all the candidate class logits.

This loss can be optionally added to semi-supervised setting, where the $p_{\text{ref}}$ could be defined as the predicted label distribution on the labeled training data $\mathbb{E}_{x \sim p_{\text{data}}}[D(x)]$.

## D    INCEPTION SCORE

As a recently proposed metric for evaluating the performance of the generative models, the Inception-Score has been found well correlated with human evaluation (Salimans et al., 2016), where a pre-trained publicly-available Inception model $C$ is introduced. By applying the Inception model to each generated sample $x$ and getting the corresponding class probability distribution $C(x)$, Inception Score is calculated via

$$\text{Inception Score} = \exp\left(\mathbb{E}_x\left[\text{KL}\left(C(x) \parallel \bar{C}^G\right)\right]\right), \tag{36}$$

where $\mathbb{E}_x$ is short of $\mathbb{E}_{x \sim G}$ and $\bar{C}^G = \mathbb{E}_x[C(x)]$ is the overall probability distribution of the generated samples over classes, which is judged by $C$, and KL denotes the Kullback-Leibler divergence which is defined as

$$\text{KL}(p \parallel q) = \sum_i p_i \log \frac{p_i}{q_i} = \sum_i p_i \log p_i - \sum_i p_i \log q_i = -H(p) + H(p,q). \tag{37}$$

An extended metric, the Mode Score, is proposed in Che et al. (2016) to take the prior distribution of the labels into account, which is calculated via

$$\text{Mode Score} = \exp\left(\mathbb{E}_x\left[\text{KL}\left(C(x) \parallel \bar{C}^{\text{train}}\right)\right] - \text{KL}\left(\bar{C}^G \parallel \bar{C}^{\text{train}}\right)\right), \tag{38}$$

where the overall class distribution from the training data $\bar{C}^{\text{train}}$ has been added as a reference. We show in the following that, in fact, Mode Score and Inception Score are equivalent.

**Lemma 3.** *Let $p(x)$ be the class probability distribution of the sample $x$, and $\bar{p}$ denote another probability distribution, then*

$$\mathbb{E}_x\left[H\left(p(x), \bar{p}\right)\right] = H\left(\mathbb{E}_x\left[p(x)\right], \bar{p}\right). \tag{39}$$

With Lemma 3, we have

$$
\begin{aligned}
\log(&\text{Inception Score}) \\
&= \mathbb{E}_x\left[\text{KL}(C(x) \parallel \bar{C}^G)\right] \\
&= \mathbb{E}_x\left[H\left(C(x), \bar{C}^G\right)\right] - \mathbb{E}_x\left[H\left(C(x)\right)\right] \\
&= H\left(\mathbb{E}_x\left[C(x)\right], \bar{C}^G\right) - \mathbb{E}_x\left[H\left(C(x)\right)\right] \\
&= H(\bar{C}^G) + \left(-\mathbb{E}_x\left[H\left(C(x)\right)\right]\right),
\end{aligned}
\tag{40}
$$

$$
\begin{aligned}
\log(&\text{Mode Score}) \\
&= \mathbb{E}_x\left[\text{KL}\left(C(x) \parallel \bar{C}^{\text{train}}\right)\right] - \text{KL}(\bar{C}^G \parallel \bar{C}^{\text{train}}) \\
&= \mathbb{E}_x\left[H\left(C(x), \bar{C}^{\text{train}}\right)\right] - \mathbb{E}_x\left[H\left(C(x)\right)\right] - H(\bar{C}^G, \bar{C}^{\text{train}}) + H(\bar{C}^G) \\
&= H(\bar{C}^G) + \left(-\mathbb{E}_x\left[H\left(C(x)\right)\right]\right).
\end{aligned}
\tag{41}
$$

# E   THE LEMMA AND PROOFS

**Lemma 1.** *With $l$ being the logits vector and $\sigma$ being the softmax function, let $\sigma(l)$ be the current softmax probability distribution and $\hat{p}$ denote any target probability distribution, then:*

$$-\frac{\partial H\big(\hat{p}, \sigma(l)\big)}{\partial l} = \hat{p} - \sigma(l). \tag{42}$$

*Proof.*

$$-\left(\frac{\partial H\big(\hat{p}, \sigma(l)\big)}{\partial l}\right)_k = -\frac{\partial H\big(\hat{p}, \sigma(l)\big)}{\partial l_k} = \frac{\partial \sum_i \hat{p}_i \log \sigma(l)_i}{\partial l_k} = \frac{\partial \sum_i \hat{p}_i \log \frac{\exp(l_i)}{\sum_j \exp(l_j)}}{\partial l_k}$$

$$= \frac{\partial \sum_i \hat{p}_i \big(l_i - \log \sum_j \exp(l_j)\big)}{\partial l_k} = \frac{\partial \sum_i \hat{p}_i l_i}{\partial l_k} - \frac{\partial \log\big(\sum_j \exp(l_j)\big)}{\partial l_k} = \hat{p}_k - \frac{\exp(l_k)}{\sum_j \exp(l_j)}$$

$$\Rightarrow -\frac{\partial H\big(\hat{p}, \sigma(l)\big)}{\partial l} = \hat{p} - \sigma(l). \qquad \square$$

**Lemma 2.** *Given $v = [v_1, \ldots, v_{K+1}]$, $v_{1:K} \triangleq [v_1, \ldots, v_K]$, $v_r \triangleq \sum_{k=1}^{K} v_k$, $R(v) \triangleq v_{1:K}/v_r$ and $F(v) \triangleq [v_r, v_{K+1}]$, let $\hat{p} = [\hat{p}_1, \ldots, \hat{p}_{K+1}]$, $p = [p_1, \ldots, p_{K+1}]$, then we have:*

$$H\big(\hat{p}, p\big) = \hat{p}_r H\big(R(\hat{p}), R(p)\big) + H\big(F(\hat{p}), F(p)\big). \tag{43}$$

*Proof.*

$$H(\hat{p}, p) = -\sum_{k=1}^{K} \hat{p}_k \log p_k - \hat{p}_{K+1} \log p_{K+1} = -\hat{p}_r \sum_{k=1}^{K} \frac{\hat{p}_k}{\hat{p}_r} \log\big(\frac{p_k}{p_r} p_r\big) - \hat{p}_{K+1} \log p_{K+1}$$

$$= -\hat{p}_r \sum_{k=1}^{K} \frac{\hat{p}_k}{\hat{p}_r} \big(\log \frac{p_k}{p_r} + \log p_r\big) - \hat{p}_{K+1} \log p_{K+1}$$

$$= -\hat{p}_r \sum_{k=1}^{K} \frac{\hat{p}_k}{\hat{p}_r} \log \frac{p_k}{p_r} - \hat{p}_r \log p_r - \hat{p}_{K+1} \log p_{K+1}$$

$$= \hat{p}_r H\big(R(\hat{p}), R(p)\big) + H\big(F(\hat{p}), F(p)\big). \qquad \square$$

**Lemma 3.** *Let $p(x)$ be the class probability distribution of the sample $x$ that from a certain data distribution, and $\bar{p}$ denote the reference probability distribution, then*

$$\mathbb{E}_x\big[H\big(p(x), \bar{p}\big)\big] = H\big(\mathbb{E}_x\big[p(x)\big], \bar{p}\big). \tag{44}$$

*Proof.*

$$\mathbb{E}_x\big[H\big(p(x), \bar{p}\big)\big] = \mathbb{E}_x\big[-\sum_i p_i(x) \log \bar{p}_i\big] = -\sum_i \mathbb{E}_x[p_i(x)] \log \bar{p}_i$$

$$= -\sum_i \big(\mathbb{E}_x[p(x)]\big)_i \log \bar{p}_i = H\big(\mathbb{E}_x\big[p(x)\big], \bar{p}\big). \qquad \square$$

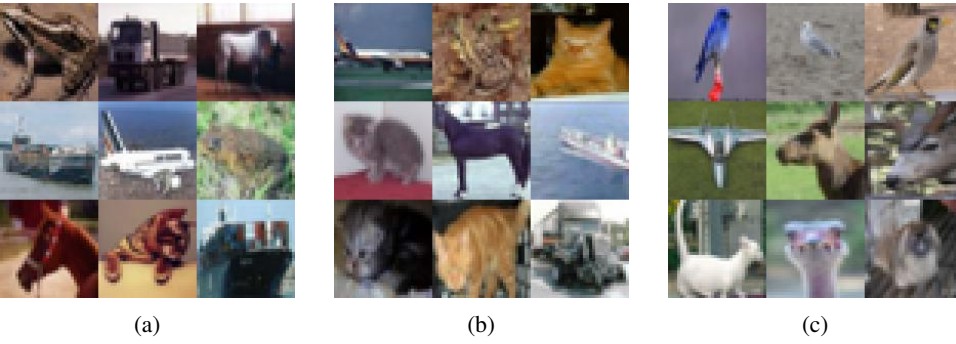

|  (a)  |  (b)  |  (c)  |

Figure 8: $H(C(x))$ of Inception Score in Real Images. a) $0<H(C(x))<1$ ; b) $3<H(C(x))<4$; c) $6<H(C(x))<7$.

# F    NETWORK STRUCTURE & HYPER-PARAMETERS

Generator:

| Operation | Kernel | Strides | Output Dims | Output Drop | Activation | BN |
|---|---|---|---|---|---|---|
| Noise | N/A | N/A | 100 \| 110 | 0.0 | N(0.0, 1.0) | |
| Linear | N/A | N/A | 4×4×768 | 0.0 | Leaky ReLU | True |
| Deconvolution | 3×3 | 2×2 | 8×8×384 | 0.0 | Leaky ReLU | True |
| Deconvolution | 3×3 | 2×2 | 16×16×192 | 0.0 | Leaky ReLU | True |
| Deconvolution | 3×3 | 2×2 | 32×32×96 | 0.0 | Leaky ReLU | True |
| Deconvolution | 3×3 | 1×1 | 32×32×3 | 0.0 | Tanh | |

Discriminator:

| Operation | Kernel | Strides | Output Dims | Output Drop | Activation | |
|---|---|---|---|---|---|---|
| Add Gaussian Noise | N/A | N/A | 32×32×3 | 0.0 | N(0.0, 0.1) | |
| Convolution | 3×3 | 1×1 | 32×32×64 | 0.3 | Leaky ReLU | True |
| Convolution | 3×3 | 2×2 | 16×16×128 | 0.3 | Leaky ReLU | True |
| Convolution | 3×3 | 2×2 | 8×8×256 | 0.3 | Leaky ReLU | True |
| Convolution | 3×3 | 2×2 | 4×4×512 | 0.3 | Leaky ReLU | True |
| Convolution* | 3×3 | 1×1 | 4×4×512 | 0.3 | Leaky ReLU | True |
| AvgPool | N/A | N/A | 1×1×512 | 0.3 | N/A | |
| Linear | N/A | N/A | 10 \| 11 \| 12 | 0.0 | Softmax | |

| The *layer was only used for class condition experiments |
|---|
| Optimizer: Adam with beta1=0.5, beta2=0.999; Batch size=100. |
| Learning rate: Exponential decay with stair, initial learning rate 0.0004. |
| We use weight normalization for each weight |

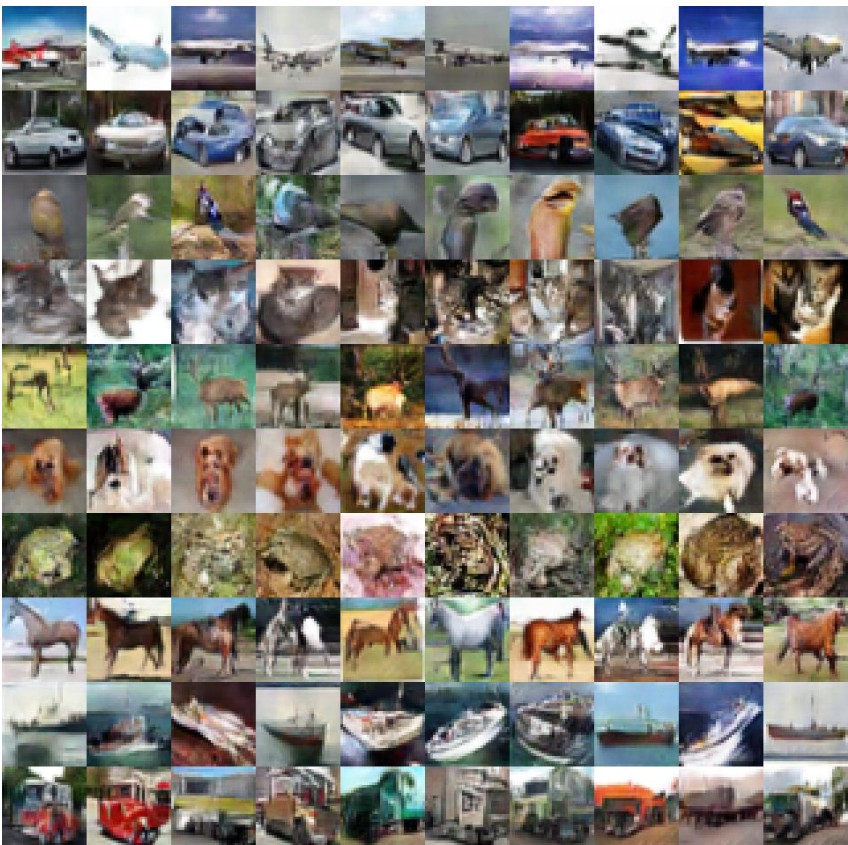

Figure 9: Random Samples of AM-GAN: Dynamic Labeling

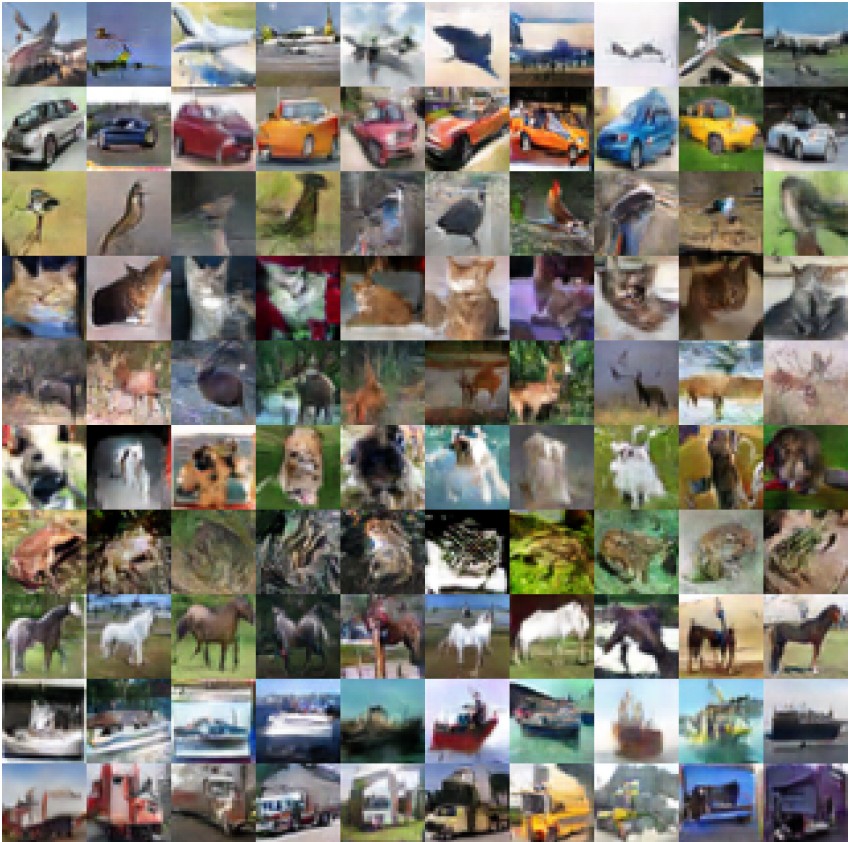

Figure 10: Random Samples of AM-GAN: Class Condition

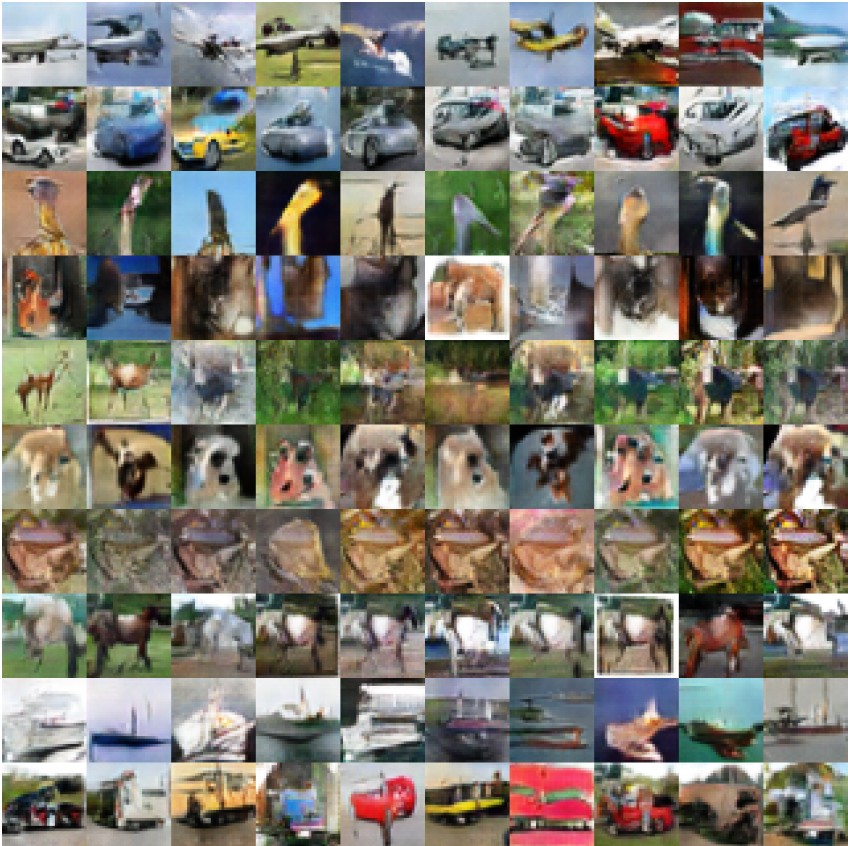

Figure 11: Random Samples of AC-GAN$^*$: Dynamic Labeling

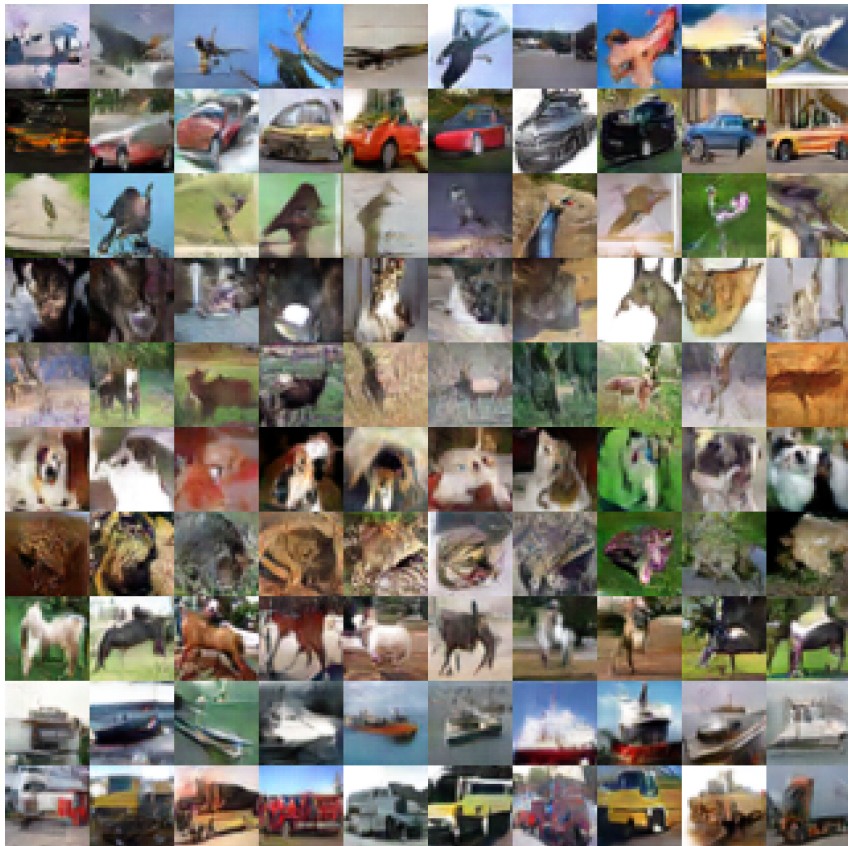

Figure 12: Random Samples of AC-GAN$^*$: Class Condition

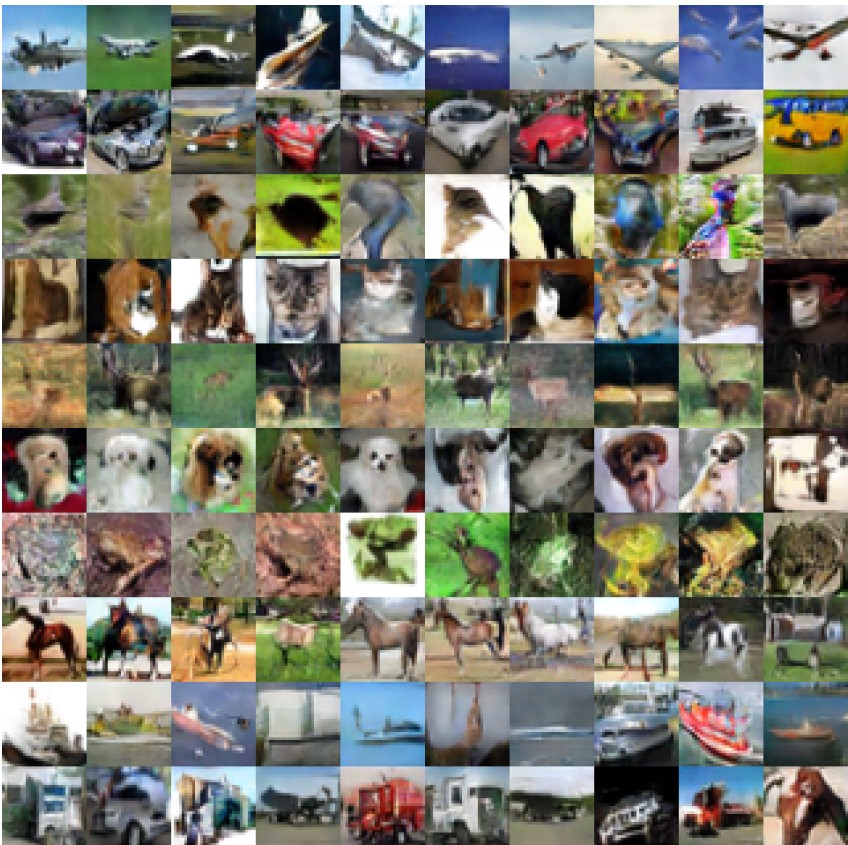

Figure 13: Random Samples of AC-GAN$^{*+}$: Dynamic Labeling

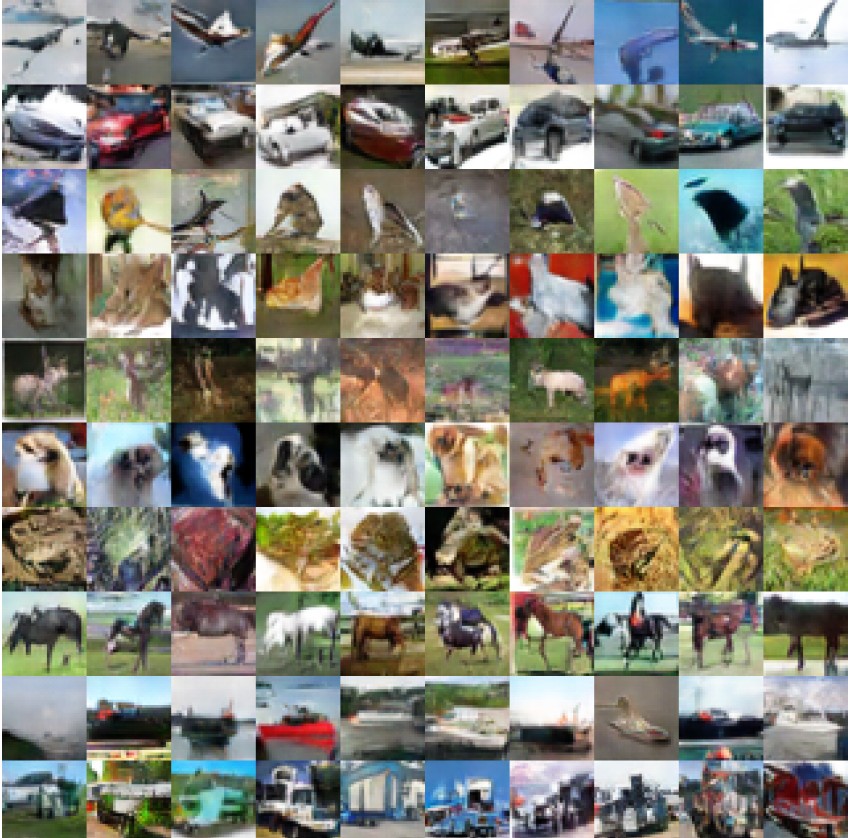

Figure 14: Random Samples of AC-GAN$^{*+}$: Class Condition

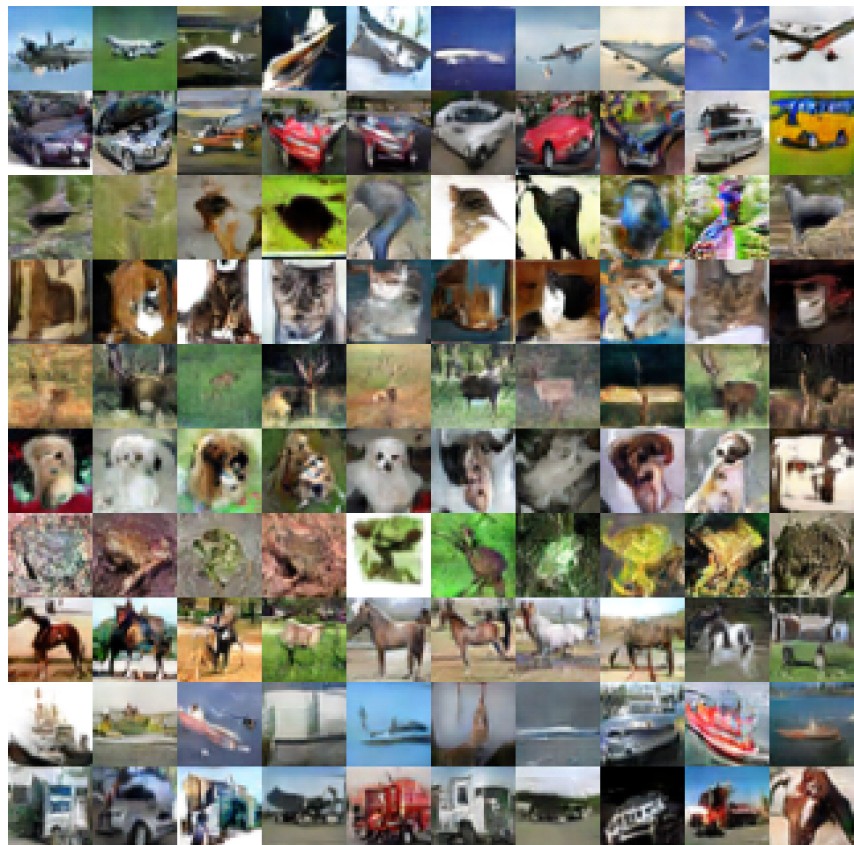

Figure 15: Random Samples of LabelGAN: Under Dynamic Labeling Setting

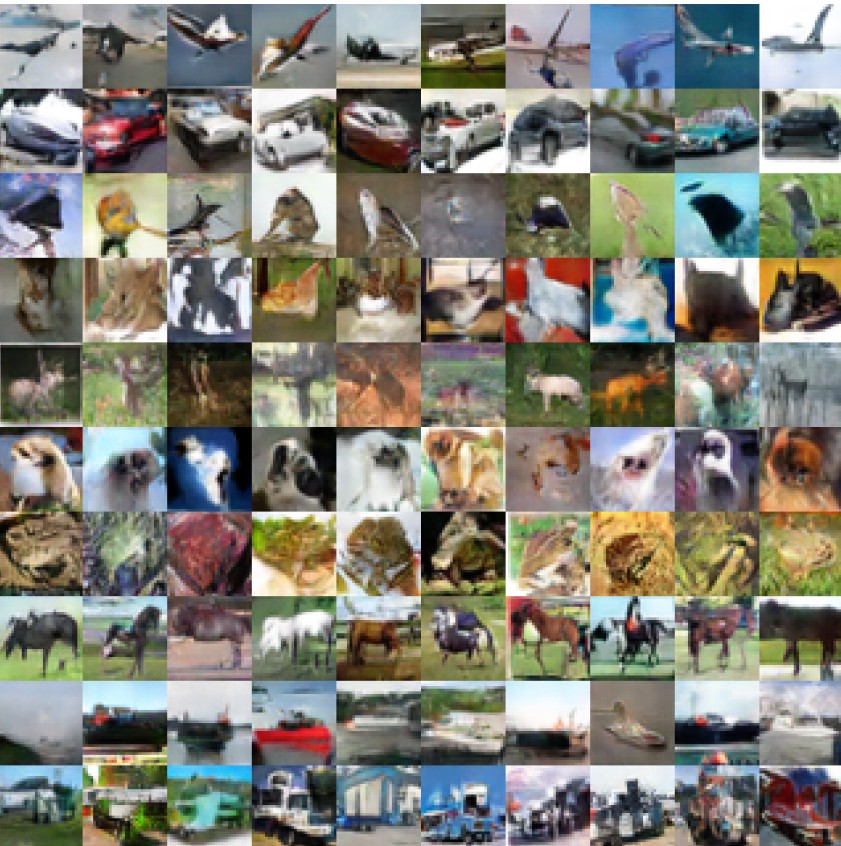

Figure 16: Random Samples of LabelGAN: Under Class Condition Setting

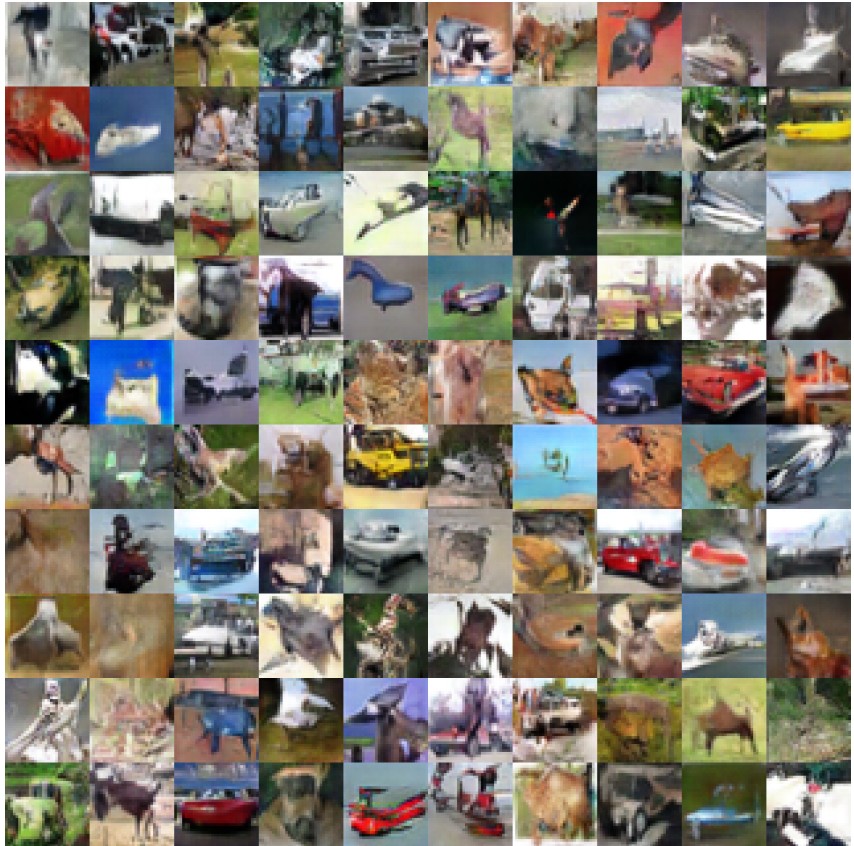

Figure 17: Random Samples of GAN: Under Dynamic Labeling Setting

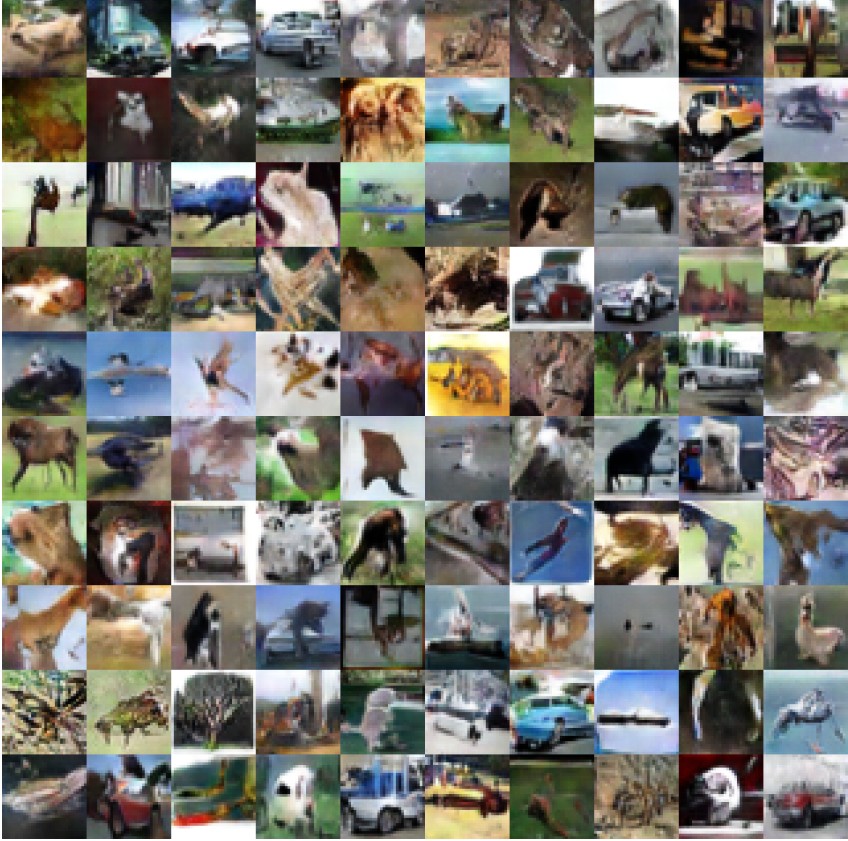

Figure 18: Random Samples of GAN: Under Class Condition Setting

