# OpenReview forum: "Activation Maximization Generative Adversarial Nets"
_ICLR.cc/2018/Conference — Accept (Poster)_

### Official Review · AnonReviewer3 · 2017-11-17
**Review of Activation Maximization**

**Rating:** 5
**Confidence:** 4

**Review:**


I thank the authors for the thoughtful responses and updated manuscript. Although the manuscript is improved, I still feel it is unfocused and may be substantially improved, thus my review score remains unchanged.

===============

The authors describe a new version of a generative adversarial network (GAN) for generating images that is heavily related to class-conditional GAN's. The authors highlight several additional results on evaluation metrics and demonstrate some favorable analyses using their new proposed GAN.

Major comments:
1) Unfocused presentation. The paper presents a superfluous and extended background section that needs to be cut down substantially. The authors should aim for a concise presentation of their work in 8 pages. Additionally, the authors present several results (e.g. Section 5.1 on dynamic labeling, Section 6.1 on Inception score) that do not appear to improve the results of the paper, but merely provide commentary. The authors should either defend why these sections are useful or central to the arguments in the paper; otherwise, remove them.

2) Quantitative evaluation highlight small gains. The gains in Table 1 seem to be quite small and additionally there are no error bars so it is hard to assess what is statistically meaningful. Table 2 highlights some error bars but again the gains some quite small. Given that the AM-GAN seems like a small change from an AC-GAN model, I am not convinced there is much gained using this model.

3) MS-SSIM. The authors' discussion of MS-SSIM is fairly confusing. MS-SSIM is a measure of image similarity between a pair of images. However, the authors quote an MS-SSIM for various GAN models in Table 3. What does this number mean?  I suspect the authors are calculating some cumululative statistics across many images, but I was not able to find a description, nor understand what these statistics mean.

4) 'Inception score as a diversity measurement.' This argument is not clear to me. Inception scores can be quite high for an individual image indicating that the image 'looks' like a given class in a discriminative model.  If a generative model always generates a single, good image of a 'dog', then the classification score would be quite high but the generative model would be very poor because the images are not diverse. Hence, I do not see how the inception score captures this property.

If the authors can address all of these points in a substantive manner, I would consider raising my rating.

---

> ### Author Response · Authors · 2017-12-06
> **Response**
>
> We sincerely thank you for your constructive advice on our paper. We have substantially revised our paper according to your comments.
>
> 1. Unfocused Presentation:
>
> a) Superfluous Preliminary. We shorten the preliminary section and only keep necessary equations that are referred in later sections.
>
> b) Section 6.1 on Inception Score. We have discarded the inessential part of the discussions in Section 6.1 and only kept the definition of Inception Score.
>
> c) Section 5.1 on Dynamic Labeling. Dynamic labeling brings important improvements to AM-GAN, and is applicable to other models that require target class for each generated sample, such as AC-GAN. It is an alternative to predefined labeling, and affects the models largely according to our experiments. As such, we consider it necessary to keep the discussions about dynamic labeling. We have made the point clearer in the revised version.
>
> 2. Quantitative Evaluation:
>
> a) No Error Bar in Table 1. We have added error bars in Table 1, where AM-GAN still consistently outperforms variants of AC-GAN and LabelGAN by a large margin in terms of both Inception Score and AM Score.
>
> b) Table 2 Shows Small Gains. As shown in Table 2, AM-GAN achieves 8.91±0.11 Inception Score on CIFAR-10, which significantly outperforms all the baseline methods, including Improved GAN (8.09±0.07), AC-GAN (8.25±0.07), WGAN-GP + AC (8.42±0.10) and SGAN (8.59±0.11). When compared to Splitting GAN (8.87±0.09), an orthogonal work to AM-GAN, which enhances the class label information via class splitting, the improvement seems to be less significant. However, since it is orthogonal to AM-GAN, we can combine them to further improve the results. As it takes some time to conduct additional experiments, we would add such results in the later version.
>
> 3. MS-SSIM:
>
> Sorry for missing the descriptions on MS-SSIM. We actually borrow the usage of MS-SSIM from AC-GAN (Odena et al., 2016) which measures the MS-SSIM scores between a set of randomly sampled pairs of images within a given class and uses the mean of the MS-SSIM scores, where a high mean MS-SSIM indicates intra-class mode collapse or low sample diversity in the class. In our paper, we report the maximum of the mean MS-SSIM over the 10 classes in CIFAR-10, with which we judge whether there exists obvious intra-class mode collapse. We have added corresponding descriptions and citations to MS-SSIM in the new version (the first paragraph in Section 7 and the caption of Table 2).
>
> 4. Inception Score as a Diversity Measurement:
>
> A direct answer is that Inception Score, i.e. exp(H(E_x[C(x)])-E_x[H(C(x))]), has two terms and, more importantly, when the generator collapsed to a single point, the first term H(E_x[C(x)]) (the entropy of “the mean classification distribution of all samples”) and the second term E_x[H(C(x))] (the mean classification entropy score for each sample) in Inception Score would be actually equal. Though the second term can be high, the overall Inception Score, in this case, would always be the minimal value 1.0 (exp^0).
>
> It is worth noting that the first and second terms are highly correlated. As in Section 6.3, we provide an alternative explanation by understanding Inception Score in the KL divergence formulation, i.e. exp(E_x[KL(C(x),E_x[C(x)])]), which involves a single term and can be interpreted as it requires each sample’s distribution to be highly different from the overall distribution of the generator. In this view, it measures the sample diversity. We further demonstrate that Inception Score can capture sample diversity well with synthetic experiments:  assuming the generator perfectly generates a subset of the training data, with the subset growing to cover the entire dataset, Inception Score is monotonically increasing. Please also refer to Sections 6.2 and 6.3 in our paper for more details.
>
> The comments have been very useful for us to improve our paper, and we have updated our paper according to your valuable review comments. Please check it.

---

> > ### Author Response · Authors · 2018-01-05
> > **Experiments on Class-Splitting**
> >
> > We have conducted additional experiments on combining AM-GAN with the class-splitting technique proposed in [1] which is orthogonal to our work and has shown to improve quality of generated samples. Unfortunately, we found that it fails to further improve Inception Score in our setting. This might be due to the fact that the quality of split-classes is not good enough, which largely depends on the features it learns and the clustering algorithm it uses. It requires further investigations to make split-classes more effective and we would leave it as future work.
> >
> > [1] Guillermo, L. Grinblat, Lucas, C. Uzal, and Pablo, M. Granitto. Class-splitting generative adversarial
> > networks. arXiv preprint arXiv:1709.07359, 2017.

---

### Official Review · AnonReviewer2 · 2017-11-23
**good paper with thorough experiments!**

**Rating:** 7
**Confidence:** 4

**Review:**

+ Pros:
- The paper properly compares and discusses the connection between AM-GAN and class conditional GANs in the literature (AC-GAN, LabelGAN)
- The experiments are thorough
- Relation to activation maximization in neural visualization is also properly mentioned
- The authors publish code and honestly share that they could not reproduce AC-GAN's results and thus using to its best variant AC-GAN* that they come up with. I find this an important practice worth encouraging!
- The analysis of Inception score is sound.
+ Cons:
- A few presentation/clarity issues as below
- This paper leaves me wonder why AM-GAN rather than simply characterizing D as a 2K-way classifier (1K real vs 1K fake).

+ Clarity:
The paper is generally well-written. However, there are a few places that can be improved:
- In 2.2, the authors mentioned "In fact, the above formulation is a modified version of the original AC-GAN..", which puts readers confusion whether they were previously just discussed AC-GAN or AC-GAN* (because the previous paragraph says "AC-GAN are defined as..".
- Fig. 2: it's not clear what the authors trying to say if looking at only figures and caption. I'd suggest describe more in the caption and follow the concept figure in Odena et al. 2016.
- A few typos here and there e.g. "[a]n diversity measurement"

+ Originality: AM-GAN is an incremental work by applying AM to GAN. However, I have no problems with this.
+ Significance:
- Authors show that in quantitative measures, AM-GAN is better than existing GANs on CIFAR-10 / TinyImageNet. Although I don't find much a real difference by visually comparing of samples of AM-GAN to AC-GAN*.

Overall, this is a good paper with thorough experiments supporting their findings regarding AM-GAN and Inception score!

---

> ### Author Response · Authors · 2017-12-06
> **Response**
>
> We sincerely thank you for your constructive feedback. We have revised the paper and fixed the confusing statements. More descriptions about the tables and figures have been added in their captions.
>
> 1. Why not characterize the discriminator as a 2K-way classifier (K real vs K fake)?
>
> This is an interesting idea and we have thought about this originally. However, we did not feel strongly that considering K fake logits would help in our case:
>
> a) Introducing specific real class logits in the discriminator makes it possible to assign a specific target class for each generated sample, which provides a clearer guidance to the generator. However, a fake class will not be used as the target for the generated sample. In this sense, how and whether we can benefit from using K specific fake class logits are still unknown.
>
> b) Introducing more fake classes does influence the gradient that the generator receives from the discriminator. When optimizing a generated sample, only the target class logit is encouraged while all the others are otherwise discouraged. Thus, replacing a single fake class with K fake classes changes the discouraged recipient from the overall fake class to the K specific fake classes. It requires further investigations to figure out whether this will help or not. We leave it as our future work.

---

### Official Review · AnonReviewer1 · 2017-11-27
**Thorough investigation and extension of class-aware GAN approaches**

**Rating:** 8
**Confidence:** 4

**Review:**

This paper is a thorough investigation of various “class aware” GAN architectures. It purposes a variety of modifications on existing approaches and additionally provides extensive analysis of the commonly used Inception Score evaluation metric.

The paper starts by introducing and analyzing two previous class aware GANs - a variant of the Improved GAN architecture used for semi-supervised results (named Label GAN in this work) and AC-GAN, which augments the standard discriminator with an auxiliary classifier to classify both real and generated samples as specific classes.

The paper then discusses the differences between these two approaches and analyzes the loss functions and their corresponding gradients. Label GAN’s loss encourages the generator to assign all probability mass cumulatively across the k-different label classes while the discriminator tries to assign all probability mass to the k+1th output corresponding to a “generated” class. The paper views the generators loss as a form of implicit class target loss.

This analysis motivates the paper’s proposed extension, called Activation Maximization. It corresponds to a variant of Label GAN where the generator is encouraged to maximize the probability of a specific class for every sample instead of just the cumulative probability assigned to label classes. The proposed approach performs strongly according to inception score on CIFAR-10 and includes additional experiments on Tiny Imagenet to further increase confidence in the results.

A discussion throughout the paper involves dealing with the issue of mode collapse - a problem plaguing standard GAN variants. In particular the paper discusses how variants of class conditioning effect this problem. The paper presents a useful experimental finding - dynamic labeling, where targets are assigned based on whatever the discriminator thinks is the most likely label, helps prevent mode collapse compared to the predefined assignment approach used in AC-GAN / standard class conditioning.

I am unclear how exactly predefined vs dynamic labeling is applied in the case of the Label GAN results in Table 1. The definition of dynamic labeling is specific to the generator as I interpreted it. But Label GAN includes no class specific loss for the generator. I assume it refers to the form of generator - whether it is class conditional or not - even though it would have no explicit loss for the class conditional version. It would be nice if the authors could clarify the details of this setup.

The paper additionally performs a thorough investigation of the inception score and proposes a new metric the AM score. Through analysis of the behavior of the inception score has been lacking so this is an important contribution as well.

As a reader, I found this paper to be thorough, honest, and thoughtful. It is a strong contribution to the “class aware” GAN literature.

---

> ### Author Response · Authors · 2017-12-06
> **Response**
>
> We sincerely thank you for your comprehensive comments on our paper.
>
> 1. What does dynamic/predefined labeling in Table 1 means? How does it apply to LabelGAN?
>
> Models with dynamic labeling and predefined labeling settings require different network structures (G and D’s capacities). The “dynamic” and “predefined” in Tables 1 and 3 represent two experimental settings which differ in network structures. Under the “dynamic” setting, if a model requires specific target class (AC-GAN, AM-GAN), we apply dynamic labeling; under the “predefined” setting, if a model requires specific target class, we apply predefined labeling; for models that do not need target class (GAN, LabelGAN), neither of them are applied and the two (“dynamic” and “predefined”) only differ in network structures. In this way, we compare various models with almost identical network structures. We have revised the caption of Table 1. It should be clear now.

---

### Public Comment · (anonymous) · 2017-11-03
**Minor comment**

In Table 2, the citation for SGAN should be Huang et al. instead.

---

> ### Author Response · Authors · 2017-11-05
> **Thanks for your correction.**
>
> Yeah, it is indeed a mistake. We will correct it in the revision. Thanks. ^_^

---

### Public Comment · (anonymous) · 2017-11-15
**Fréchet Inception Distance (FID) to evaluate GANs**

[1] proposed the Fréchet Inception Distance (FID) to evaluate GANs which is the Fréchet distance aka Wasserstein-2 distance between the real world and generated samples statistics.  The statistics consists of the first two moments such that the sample quality (first moment match) and variation (second moment match) are covered.

As highlighted here in Section 6.2, for datasets not covering all ImageNet classes e.g. celebA, CIFAR-10 etc, the entropy of E_x~G[C(x)] is going down not up as soon as a trained GAN starts producing correctly samples falling only in some of the ImageNet classes.  [1] also showed inconistent behaviour of the Inception Score in their experiments (see Appendix A1). Especially interesting here is experiment 6 where a dataset (celebA) is increasingly mixed with ImageNet samples. The Inception Score shows a contradictory behaviour, while the FID captures this contamination, and other disturbance variants, very well.

The authors should discuss their proposed AM Score compared to the FID, also under consideration that the FID does not
need an accordingly pretrained classifier.

[1] https://arxiv.org/abs/1706.08500

---

> ### Public Comment · (anonymous) · 2017-11-27
> **Discussion on Fréchet Inception Distance (FID)**
>
> Thanks for your reference to Fréchet Inception Distance (FID).
>
> FID measures the distance between two distributions using their means and variances after a fixed mapping, e.g. Inception Network, which works well in practice as illustrated in [1]. As another evaluation metric for generative models, we would add discussions in the revision.
>
> A concern on FID is that the mean and variance of the distribution are not sufficient to represent the whole distribution. That is, for any given distribution, we can always design another distribution which is totally different from the given distribution but has the same mean and variance. We are not sure whether this would cause a problem in practice.
>
> Also, FID is actually orthogonal to Inception Score and AM Score. FID directly measures the distance between generated distribution and real-data distribution, while Inception Score mainly measures the sample diversity and AM Score mainly measures the sample quality.
>
> As for the failure of Inception Score on CelebA illustrated in [1], according to our analysis, Inception Score works as a diversity measurement and we might need a more suitable classifier (maybe a classifier trained on a face dataset) to make it work on CelebA. FID seems to have the benefit of being not sensitive to the choice of the mapping function, though it also remains uncertain whether the Inception Network is always the best choice as the mapping function for variant models.

---

### Decision · Program_Chairs · 2018-01-29
**ICLR 2018 Conference Acceptance Decision**

**Decision:**

Accept (Poster)

**Comment:**

The authors investigate various class aware GANs and provide extensive analysis of their ability to address mode collapse and sample quality issues. Based on this analysis they propose an extension called Activation Maximization-GAN which tries to push each generated sample to a specific class indicated by the Discriminator. As experiments show, this leads to better sample quality & helps with mode collapse issue. The authors also analyze inception score to measure sample quality and propose a new metric better suited for this task.